# Absence of CEP78 causes photoreceptor and sperm flagella impairments in mice and a human individual

Tianyu Zhu[1†], Yuxin Zhang[2,3†], Xunlun Sheng[4,5†], Xiangzheng Zhang[1], Yu Chen[1], Hongjing Zhu[3], Yueshuai Guo[1], Yaling Qi[1], Yichen Zhao[1], Qi Zhou[1], Xue Chen[3*], Xuejiang Guo[1*], Chen Zhao[2*]

[1]State Key Laboratory of Reproductive Medicine, Department of Histology and Embryology, Gusu School, Nanjing Medical University, Nanjing, China; [2]Department of Ophthalmology and Vision Science, Eye & ENT Hospital, Shanghai Medical College, Fudan University, Shanghai, China; [3]Department of Ophthalmology, The First Affiliated Hospital of Nanjing Medical University, Nanjing Medical University, Nanjing, China; [4]Gansu Aier Ophthalmiology and Optometry Hospital, Lanzhou, China; [5]Ningxia Eye Hospital, People's Hospital of Ningxia Hui Autonomous Region, Third Clinical Medical College of Ningxia Medical University, Yinchuan, China

*For correspondence:
drcx1990@vip.163.com (XC);
guo_xuejiang@njmu.edu.cn (XG);
dr_zhaochen@163.com (CZ)

†These authors contributed equally to this work

Competing interest: The authors declare that no competing interests exist.

**Abstract** Cone-rod dystrophy (CRD) is a genetically inherited retinal disease that can be associated with male infertility, while the specific genetic mechanisms are not well known. Here, we report *CEP78* as a causative gene of a particular syndrome including CRD and male infertility with multiple morphological abnormalities of sperm flagella (MMAF) both in human and mouse. *Cep78* knockout mice exhibited impaired function and morphology of photoreceptors, typified by reduced ERG amplitudes, disrupted translocation of cone arrestin, attenuated and disorganized photoreceptor outer segments (OS) disks and widen OS bases, as well as interrupted connecting cilia elongation and abnormal structures. *Cep78* deletion also caused male infertility and MMAF, with disordered '9+2' structure and triplet microtubules in sperm flagella. Intraflagellar transport (IFT) proteins IFT20 and TTC21A are identified as interacting proteins of CEP78. Furthermore, CEP78 regulated the interaction, stability, and centriolar localization of its interacting protein. Insufficiency of CEP78 or its interacting protein causes abnormal centriole elongation and cilia shortening. Absence of CEP78 protein in human caused similar phenotypes in vision and MMAF as *Cep78⁻/⁻* mice. Collectively, our study supports the important roles of *CEP78* defects in centriole and ciliary dysfunctions and molecular pathogenesis of such multi-system syndrome.

## Editor's evaluation

This paper is of interest to scientists within the cilia and centrosome fields, especially those studying photoreceptor and sperm development and the diseases associated with their dysfunction. The phenotypic characterization Cep78-/- mutant mice, revealing severe structural and functional defects of photoreceptors and sperm flagella, is convincing and consistent with recently published work. While CEP78 is shown to physically interact with IFT20 and TTC21A, the precise molecular mechanisms by which CEP78 affects photoreceptor and sperm development remain to be clarified.

## Introduction

Cone-rod dystrophy (CRD), a genetically inherited retinal disease, features photoreceptor degeneration (*Gill et al., 2019*). A small portion of CRDs is due to retina cilia defects, and they may have cilia-related syndromic defects in other systems (*Gill et al., 2019*). In this circumstance, pathogenic genes of those CRDs are associated with connecting cilia in photoreceptor, a type of non-motile cilia (*Khanna et al., 2005*). *CEP250* mutation is related to CRD and hearing loss (CRDHL; *Kubota et al., 2018*), and *CEP19* mutation to CRD with obesity and renal malformation (*Yıldız Bölükbaşı et al., 2018*). Different from photoreceptor connecting cilia, sperm flagellum is motile cilia. Motile and immotile cilia defects usually do not occur at the same time. However, occasionally, CRD can also develop as a syndrome together with abnormalities of sperm flagella. As this syndrome leads to defects of two unrelated physiological functions of vision and reproduction, comprehensive analysis of its specific genetic etiology and molecular mechanism is still limited.

Centrosomal protein of 78 kDa (CEP78), protein encoded by the *CEP78* gene (MIM: 617110), is a centriolar protein composed of 722 amino acids and possesses two leucine-rich repeat regions and a coiled-coil domain (*Brunk et al., 2016*). CEP78 localizes to the distal region of mature centrioles and is exclusively expressed in ciliated organisms, supporting its crucial function in maintaining centrosome homeostasis (*Hossain, 2017*; *Azimzadeh et al., 2012*). Mutations in the *CEP78* gene cause autosomal recessive CRDHL (MIM: 617236; *Fu, 2017*; *Nikopoulos et al., 2016*; *Namburi et al., 2016*; *Sanchis-Juan et al., 2018*; *Ascari et al., 2020*). Deregulation of Cep78 also contributes to colorectal cancer (*Zhang et al., 2016a*), prostate cancer (*Nesslinger et al., 2007*), and asthenoteratozoospermia (*Ascari et al., 2020*). As a centrosomal protein, CEP78 functions downstream of CEP350 to control biogenesis of primary cilia (*Gonçalves et al., 2021*) and regulates centrosome homeostasis through interactions with other centrosomal proteins, including CEP76, CP110, and EDD-DYRK2-DDB1$^{VprBP}$ (*Hossain, 2017*). Despite these findings, the biological function of Cep78, especially its function and mechanism in CRD with male infertility, has not been well understood yet.

In this study, based on results of a male patient carrying *CEP78* mutation and *Cep78* gene knockout mice, we report *CEP78* as a causative gene for CRD and male sterility. We revealed that CEP78 deficiency caused dysfunction and irregular ciliary structure of photoreceptors. Deletion of *Cep78* triggered male infertility, spermatogenesis defects, aberrant sperm flagella structures and manchette formation, and abnormal spermatid centriole length in male mice. The male patient carrying homozygous *CEP78* mutation leading to CEP78 protein loss also presented with syndromic phenotypes including CRD, hearing loss, and multiple morphological abnormalities of the sperm flagella (MMAF). IFT20 and TTC21A are both IFT proteins interacting with each other and vital for spermatogenesis and cilliogenesis (*Liu et al., 2019*). To construct and maintain cilia, IFTs are recruited to the basal body and traffic proteins between cilia and the cell body (*Hibbard et al., 2021*). IFT particle is assembled with two subcomplexes (IFT-A and -B; *Wingfield et al., 2021*). IFT20 is part of IFT-B subcomplex (*Petriman et al., 2022*) while TTC21A is part of IFT-A complex (*Hesketh et al., 2022*). We found that CEP78 interacted with IFT20 and TTC21A, which are vital for spermatogenesis and cilliogenesis (*Liu et al., 2019*; *Keady et al., 2011*; *Zhang et al., 2016a*), and regulated their interaction and stability. It recruited centrosome localization of IFT20. Defects of CEP78 and its interaction protein might lead to extended centriole and shortened cilia.

## Results

### Generation of *Cep78*$^{-/-}$ mice

To investigate the role of CEP78, we generated *Cep78* knockout mice using clustered regularly interspaced short palindromic repeats (CRISPR)/Cas 9 system. A single-guide RNA (sgRNA) targeting exons 2 and 11 of the *Cep78* mice gene (*Figure 1—figure supplement 1A*) was designed, generated, and injected into C57BL/6 mice zygote with the Cas 9 RNA. Genotyping confirmed deletion of exons 2–11 in heterozygotes and homozygotes (*Figure 1—figure supplement 1B*). As revealed by immunoblotting, expression of Cep78 protein was completely absent in the neural retina and testes of *Cep78*$^{-/-}$ mice (*Figure 1—figure supplement 1C–D*), indicating successful generation of *Cep78* knockout mice.

## Photoreceptor impairments of *Cep78⁻/⁻* mice

To reveal the retinal phenotypes associated with *Cep78* deletion, we studied the functional and morphological changes in *Cep78⁻/⁻* mice retina. Initially, we used ERG to detect visual functions of *Cep78⁻/⁻* mice. ERG results of *Cep78⁻/⁻* mice and their heterozygous littermates at ages of 3, 6, 9, and 18 months were recorded. Scotopic a- and b-wave and photopic b-wave amplitudes were reduced in *Cep78⁻/⁻* mice when compared to age-matched *Cep78⁺/⁻* mice (*Figure 1A–E*). The reduction extended as age increased. According to our data, amplitudes of scotopic a-wave showed approximately 13.6, 39.1, 40.8, and 64.8% decrease in *Cep78⁻/⁻* mice at 3, 6, 9, and 18 months, respectively (*Figure 1A and C*), and amplitudes of scotopic b-wave were downregulated by 19.9, 30.5, 35.6, and 58.3% in *Cep78⁻/⁻* mice at 3, 6, 9, and 18 months, respectively (*Figure 1A and D*). Similarly, amplitudes of photopic b-wave were decreased by 39.4, 39.8, 52.8, and 69.0% in *Cep78⁻/⁻* mice at 3, 6, 9, and 18 months, respectively (*Figure 1B and E*).

In normal retina, light exposure drives rapid movement of cone arrestin into outer segments (OS) of cone photoreceptors (*Zhu et al., 2002*). Herein, we tested whether *Cep78* knockout would disrupt such translocation of cone arrestin in cone photoreceptors. Immunostaining patterns of cone arrestin in dark- and light-adapted *Cep78⁺/⁻* and *Cep78⁻/⁻* mice retinas were compared. In dark-adapted retina, cone arrestin was located in the synaptic region, outer nuclear layer (ONL), and outer and inner segments. No difference was observed in distribution patterns of cone arrestin between *Cep78⁺/⁻* and *Cep78⁻/⁻* mice, while the fluorescence intensity of cone arrestin at light exposure was lower in *Cep78⁻/⁻* mice compared with *Cep78⁺/⁻* mice (*Figure 1F–G*). Light exposure of 2 hr triggered a shift of cone arrestin from the inner cellular compartments to the OS in both *Cep78⁺/⁻* and *Cep78⁻/⁻* mice. However, light-induced translocation of cone arrestin in *Cep78⁻/⁻* mice retina was slower and more limited than in *Cep78⁺/⁻* mice (*Figure 1F–G*). Thus, above data suggest that regular functions of retina are disturbed in *Cep78⁻/⁻* mice.

We next investigated morphological changes in retina of *Cep78⁻/⁻* mice. Spectral domain-optical CT system (SD-OCT) was applied to visualize all layers of mice retina. SD-OCT showed that thicknesses of ONL were attenuated in *Cep78⁻/⁻* mice when compared to *Cep78⁺/⁻* mice at ages of 3, 9, and 12 months (*Figure 1H–K*). The attenuation became more evident along with increase of age (*Figure 1H–K*). We also utilized transmission electron microscopy (TEM) to observe the ultra-structure of photoreceptors. As evidenced by TEM, photoreceptor OS were regularly lined and shaped in *Cep78⁺/⁻* mice at 3-, 6-, and 9-month age. However, disorganized, sparse, and caduceus photoreceptor OS disks and widened OS bases were observed in age-matched *Cep78⁻/⁻* mice (*Figure 1L*). Collectively, our results indicate that both functions and morphologies of mice photoreceptor are disturbed upon *Cep78* deletion.

## *Cep78⁻/⁻* mice exhibit disturbed connecting cilia structures in photoreceptors

Since CEP78 is a centrosome protein important for ciliogenesis and *CEP78* mutation was found to be associated with primary-cilia defects (*Nikopoulos et al., 2016*), we then aimed to analyze whether *Cep78⁻/⁻* mice exhibit abnormal ciliary structure using immunofluorescence staining and TEM. Connecting cilium were labeled with antibody against Nphp1 in immunofluorescence staining (*Figure 2A*). Our data indicate that connecting cilia are shortened in *Cep78⁻/⁻* mice at 3-, 6-, and 9-month age compared to age-matched *Cep78⁺/⁻* (*Figure 2B*). We further used TEM to visualize longitudinal sections of the ciliary region of photoreceptors. Consistent with immunofluorescence, TEM showed photoreceptors of *Cep78⁻/⁻* mice had shortened connecting cilia (*Figure 2C–D*). Moreover, we observed swelled upper part and disorganized microtubules in connecting cilia of *Cep78⁻/⁻* mice (*Figure 2E–G*). However, structures of basal bodies, adjacent daughter centrioles, or other organelles of inner segment were not altered in *Cep78⁻/⁻* mice (*Figure 2E–G*). Together, our data show that *Cep78* deficiency interrupts cilia elongation and disturbed ciliary structures in photoreceptors.

## Cep78 deletion-induced male infertility and MMAF in mice

Unexpectedly, we found that *Cep78⁻/⁻* male mice were infertile during the breeding of *Cep78⁻/⁻* mice. We thus explored the reproductive functions of *Cep78⁻/⁻* male mice. No difference was revealed in testis weight, body weight, and testis/body weight between *Cep78⁺/⁻* and *Cep78⁻/⁻* male mice (*Figure 3—figure supplement 1A-C*). Mating test with *Cep78⁺/⁻* female mice for 3 months showed

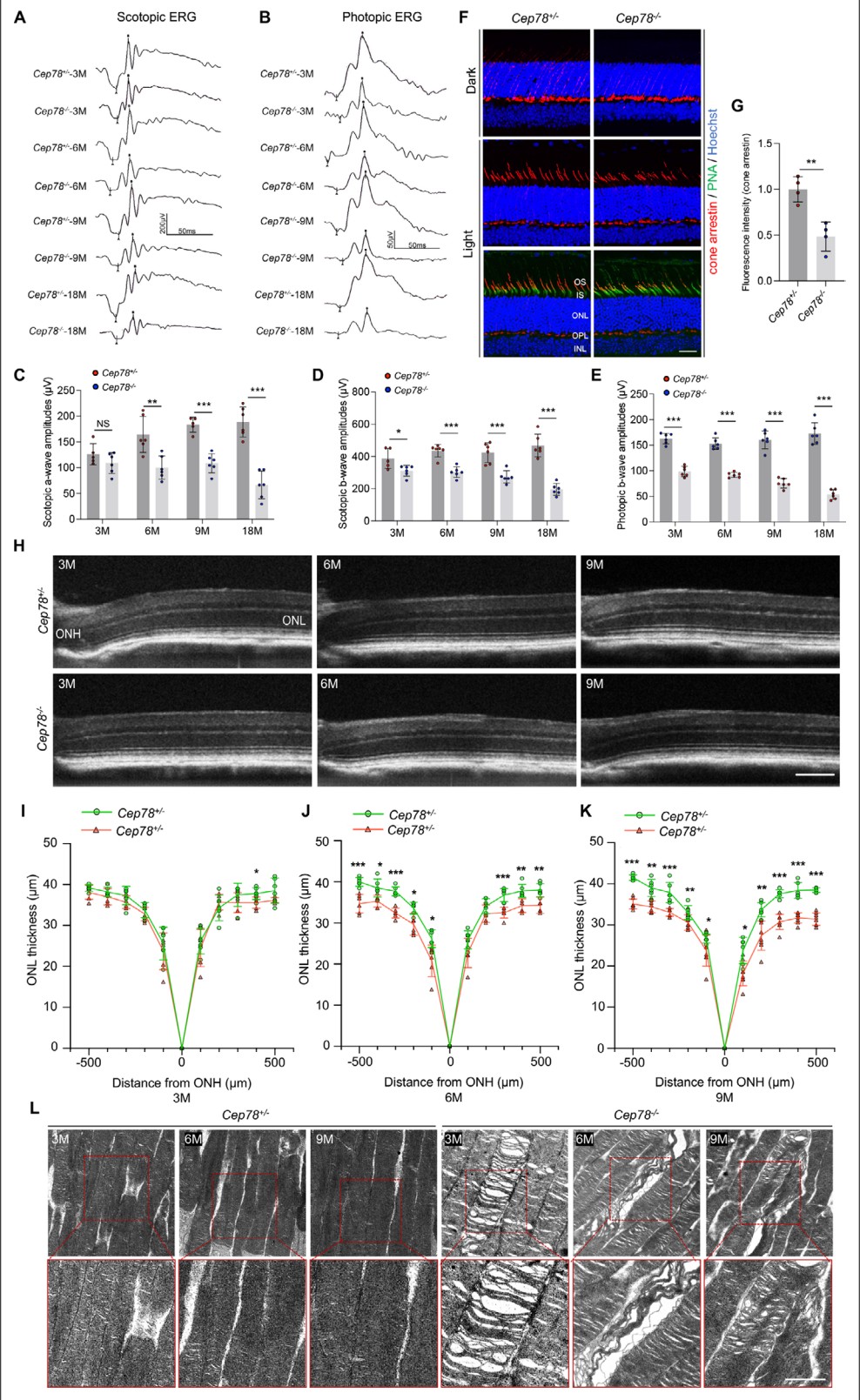

**Figure 1.** *Cep78* deletion leads to photoreceptor impairments in mice retina. (**A–E**) Scotopic and photopic ERG of *Cep78⁻ᐟ⁻* mice at the indicated ages. Representative images along with the quantification results were shown (n=6 for each sample, two-tailed Student's t-test). (**F–G**) Immunofluorescence staining of cone arrestin and peanut agglutinin (PNA) lectin in dark- and light-adapted retinas of *Cep78⁺ᐟ⁻* and *Cep78⁻ᐟ⁻* mice at 3 months. Scale bar,

*Figure 1 continued on next page*

*Figure 1 continued*

25 μm. Representative images (**F**) along with quantification results (**G**) of cone arrestin signal intensity at light exposure condition were shown (n=4 for each sample, two-tailed Student's t-test). (**H–K**) Spectral domain-optical CT (SD-OCT) revealed thickness of outer nucleic layer (ONL) in retinas of *Cep78*<sup>+/−</sup> and *Cep78*<sup>−/−</sup> mice of indicated ages. Scale bar, 150 μm. Representative images (**H**) along with the quantification results (**I–K**) were shown (n=6 for each sample, two-tailed Student's t-test). (**L**) Transmission electron microscopy (TEM) revealed ultra-structures of photoreceptors in *Cep78*<sup>+/−</sup> and *Cep78*<sup>−/−</sup> mice at the indicated ages, scale bar, 1 μm. N.S, not significant, *, p<0.05, **, p<0.01, and ***, p<0.001.

The online version of this article includes the following video, source data, and figure supplement(s) for figure 1:

**Source data 1.** Original numbers used for quantification in *Figure 1C–E, G, I, J, and K*.

**Source data 2.** Immunofluorescence, spectral domain-optical CT (SD-OCT), and transmission electron microscopy (TEM) analysis images in *Figure 1F, H and L*.

**Figure supplement 1.** Generation of *Cep78*<sup>−/−</sup> mice using the CRISPR/Cas 9 system.

**Figure supplement 1—source data 1.** Uncropped gels and blots of *Figure 1—figure supplement 1B, C, D*.

**Figure 1—video 1.** Sperm motility analysis video of *Cep78*<sup>+/−</sup> male mice.
https://elifesciences.org/articles/76157/figures#fig1video1

**Figure 1—video 2.** Sperm motility analysis video of *Cep78*<sup>−/−</sup> male mice.
https://elifesciences.org/articles/76157/figures#fig1video2

that *Cep78*<sup>−/−</sup> male mice were infertile (*Figure 3A*). To further explore the underlying causes of male infertility of *Cep78*<sup>−/−</sup> mice, we assessed the concentration, motility, and progressive motility of sperm isolated from cauda epididymis using computer-assisted sperm analysis (CASA), and all three parameters were decreased upon *Cep78* deletion (*Table 1*, *Figure 1—video 1* and *Figure 1—video 2*). Correspondingly, hematoxylin-eosin (H&E) staining also showed less spermatozoa in both caput and cauda epididymis of *Cep78*<sup>−/−</sup> male mice compared with age matched *Cep78*<sup>+/−</sup> male mice (*Figure 3B*).

Multiple abnormalities of sperm head and flagella in *Cep78*<sup>−/−</sup> male mice were initially detected by light microscopy (*Figure 3C–D*) and scanning-electron microscopy (SEM; *Figure 3E–I*), which identified severely distorted sperm heads and flagella (*Figure 3D–*), absent (*Figure 3D and F*), short (*Figure 3D and G*), coiled (*Figure 3D and H*), and/or multi-flagella (*Figure 3I*). Spermatozoa with abnormal heads, necks, and flagella accounted for 92.10, 82.61 and 95.67% of total spermatozoa, respectively, in *Cep78*<sup>−/−</sup> male mice (*Table 1*). Immunofluorescence staining was also used to study sperm morphology, results of which were consistent with those of light microscopy and SEM. Sperm axoneme was stained using antibody against Ac-α-tubulin, and short flagella were frequently observed (*Figure 3J–M*).

We isolated elongating spermatids of *Cep78*<sup>+/−</sup> and *Cep78*<sup>−/−</sup> mice using a velocity sedimentation testicular cell separation system called STA-PUT (*Bryant et al., 2013*). We next applied quantitative mass spectrometry (MS) on elongating spermatids lysates of *Cep78*<sup>+/−</sup> and *Cep78*<sup>−/−</sup> mice to reveal effects of Cep78 insufficiency on testicular protein expressions and pathway regulations. We set the threshold as a fold-change greater than 1.5 and p value less than 0.05 and identified a total of 806 downregulated proteins and 80 upregulated proteins upon *Cep78* depletion, which were visualized using hierarchical clustering analyses (*Figure 3N* and *Supplementary file 1A*). Consistent with above findings in morphological anomalies in sperm, significantly enriched gene ontology terms in cellular components included sperm principal piece, sperm fibrous sheath, acrosomal vesicle, motile cilium, centrosome, microtubule skeleton, and dynein complex (*Figure 3O* and *Supplementary file 1B*). Taken together, *Cep78* deletion decreases count, motility and progressive motility of sperm, morphological abnormalities of sperm head, and flagella in mice, as well as dysregulated sperm proteins, thus accounting for male infertility.

## Defective microtubule arrangements and elongated centrioles in sperm flagella of *Cep78*<sup>−/−</sup> mice

To determine which spermatogenesis stage that *Cep78* affected, we compared histology of testicular tissue sections between *Cep78*<sup>−/−</sup> and *Cep78*<sup>+/−</sup> male mice using periodic acid-Schiff (PAS) staining. Light microscopy showed an apparent scarcity of sperm flagella visible in the lumens of *Cep78*<sup>−/−</sup> testes (*Figure 4A–B*), with flagellar formation defects observed from stages I–III to stages VII–VIII

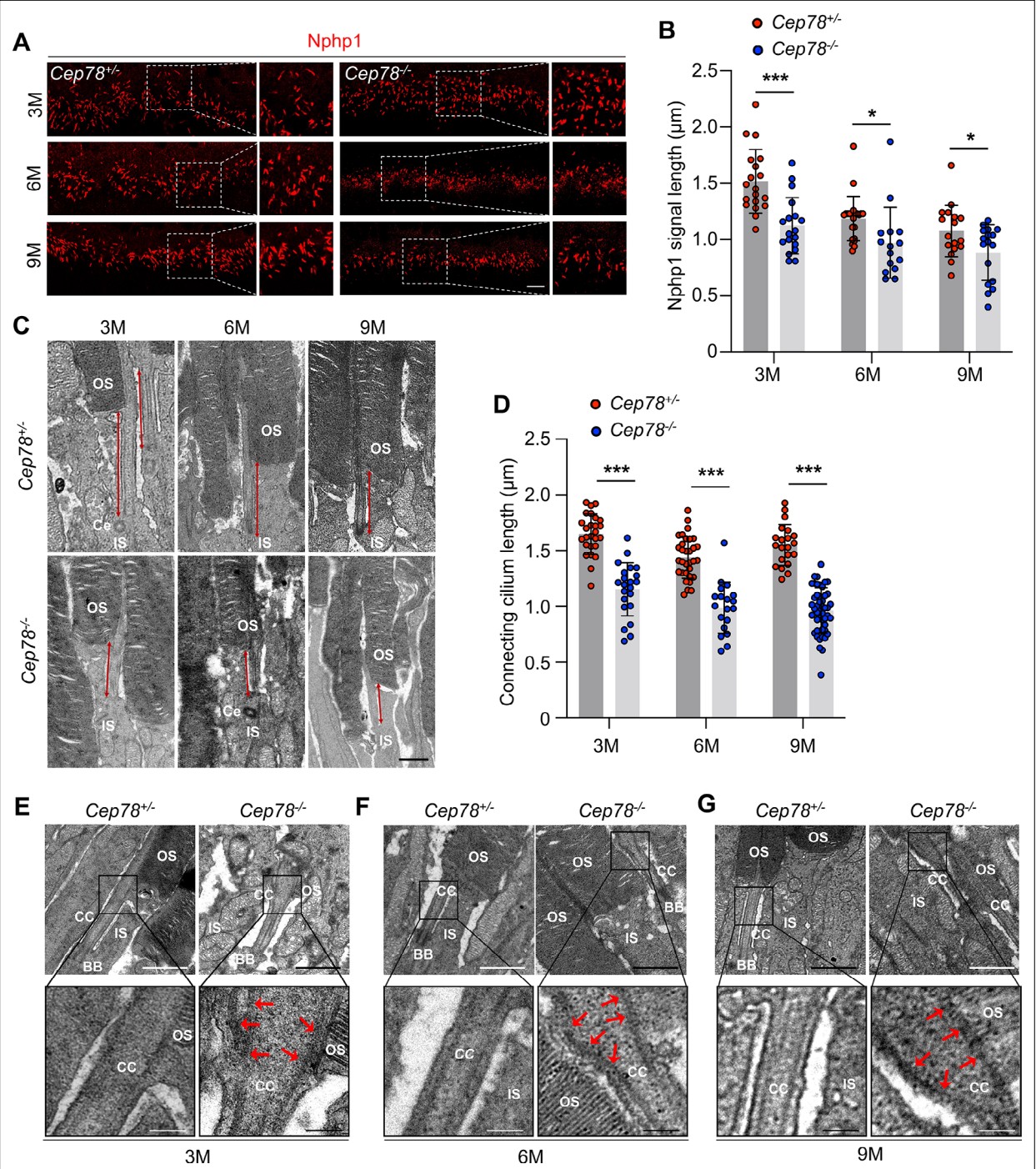

**Figure 2.** *Cep78−/−* mice exhibit disturbed ciliary structure in photoreceptors. (**A–B**) Retinal cryosections from *Cep78+/−* and *Cep78−/−* mice at indicated ages were stained with anti-Nphp1 (red) to visualize the connecting cilium (**A**) along with the quantification (**B**). Scale bar, 5 µm (three biological replications, n=19 for 3 M *Cep78+/−*, n=18 for 3 M *Cep78−/−*, n=23 for 6 M *Cep78+/−*, n=15 for 6 M *Cep78−/−*, n=17 for 9 M *Cep78+/−*, n=17 for 9 M *Cep78−/−*, two-tailed Student's t-test). (**C–D**) Transmission electron microscopy (TEM) was applied to observe longitudinal sections of the ciliary region of photoreceptors in *Cep78+/−* and *Cep78−/−* mice at indicated ages. Scale bar, 1 µm. OS, outer segment; IS, inner segment; Ce, Centriole (three biological replications, n=26 for 3 M *Cep78+/−*, n=22 for 3 M *Cep78−/−*, n=32 for 6 M *Cep78+/−*, n=19 for 6 M *Cep78−/−*, n=21 for 9 M *Cep78+/−*, n=47 for 9 M *Cep78−/−*, two-tailed Student's t-test). (**E–G**) TEM was used to observe the ultrastructure of connecting cilium in photoreceptors of *Cep78+/−* and *Cep78−/−* mice at indicated ages. Scale bar, upper 1 µm, below 200 nm. OS, outer segment; CC, connecting cilium; IS, inner segment; BB, basal body. Red arrows indicated the swelled upper part and disorganized microtubules of connecting cilia. *, p<0.05 and ***, p<0.001.

The online version of this article includes the following source data for figure 2:

*Figure 2 continued on next page*

*Figure 2 continued*

**Source data 1.** Original numbers used for quantification of *Figure 2B and D*.

**Source data 2.** Immunofluorescence and transmission electron microscopy (TEM) analysis images in *Figure 2A, C, E, F and G*.

during spermatogenesis (*Figure 4B*), suggesting that *Cep78* was involved at the stage of sperm flagellar formation. Very few spermatids reached maturity successfully and passed into epididymis (*Figure 3B* and *Table 1*).

For detailed characterization of *Cep78*$^{-/-}$ mice phenotype, we analyzed the ultra-structure of mouse testes using TEM. *Cep78*$^{+/-}$ sperm showed well-aligned arrangements of connecting segments, axoneme microtubules, mitochondria, outer dense fibers (ODFs), and fiber sheaths in longitudinal section of neck, middle, and principal pieces (*Figure 4C–D*). However, disordered mitochondria and a jumble of unassembled flagellar components were present in *Cep78*$^{-/-}$ sperm (*Figure 4E–F*). In cross sections of middle and principal regions, *Cep78*$^{+/-}$ sperm exhibited well-aligned ODF and typical '9+2' arrangement of microtubules, which were typified by nine peripheral microtubule doublets surrounding a central pair of singlet microtubules (*Figure 4G–H*). In contrast, *Cep78*$^{-/-}$ sperm showed complete derangement of ODF and scattered '9+2' microtubules (*Figure 4I–J*). In normal sperm flagella, distal centriole develops an atypical centriole structure, whose microtubules are doublets instead of triplets. Triplet microtubules can only be observed at proximal centrioles (*Avidor-Reiss and Fishman, 2019*). In *Cep78*$^{-/-}$ sperm, triplet microtubules, which were supposed to exist only in proximal centrioles, were found at both middle and principal pieces of sperm flagella (*Figure 4I–J*), indicative of abnormal elongation of centrioles during sperm flagella formation.

Tubules in axoneme are formed from centrosome. The various types of aberrant sperm flagella observed in mice with *Cep78* defects, together with the triplet microtubules found in principal and middle pieces of sperm flagella upon *Cep78* deletion, suggested centriole anomalies as potential causative factors of deregulation of sperm development in mouse. Previous in vitro studies showed that interference of *Cep78* led to elongation of centriole (*Hossain, 2017*). To further elucidate whether *Cep78* regulates centriole length in spermiogenesis, we stained centrioles and peri-centriole materials with anti-centrin 1 and anti-γ-tubulin, respectively. Two-dot staining pattern of disengaged centrioles was observed in *Cep78*$^{+/-}$ spermatids (*Figure 4K*), while depletion of *Cep78* promoted centriole elongation in spermatids (*Figure 4K*), supporting the role of *Cep78* in regulating centriole structures and functions. Collectively, our data reveal defective microtubule arrangements and elongated centrioles in sperm flagella of *Cep78*$^{+/-}$ mice.

## Abnormalities of spermatid head formation during spermiogenesis in *Cep78*$^{-/-}$ testes

In addition to axonemal defects in sperm flagella, *Cep78*$^{-/-}$ testes also exhibit deformation of spermatid nuclei, as evidenced by abnormal club-shaped nuclear morphology of elongated spermatids since step 10, in contrast to the normal hook-shaped head in *Cep78*$^{+/-}$ elongated spermatids (*Figure 4B*, *Figure 5A*). Abnormal elongation of manchette structures in testicular spermatids of *Cep78*$^{-/-}$ mice during spermiogenesis was also shown by immunofluorescence staining (*Figure 5B*), which validated above findings by PAS staining.

Further analyses with TEM observed various ultra-structural defects in sperm head formation of *Cep78*$^{-/-}$ spermatids. As indicated by TEM, *Cep78*$^{+/-}$ spermatids underwent a series of dramatic changes and showed regular reshaping of sperm head, including nuclear condensing, manchette formation, and acrosomal biogenesis (*Figure 5C, G and J*). TEM suggested that nuclear condensing was normal in *Cep78*$^{-/-}$ spermatids (*Figure 5D–F– and K–M*). Similar expressional intensity and localization of transition protein 1 from stage IX of spermatogenesis to stages I–III in *Cep78*$^{+/-}$ and *Cep78*$^{-/-}$ mice were revealed by immunofluorescence staining, confirming the TEM data, and suggesting that nuclear condensation was not affected upon *Cep78* deletion (*Figure 5N*). We also performed spermatocyte spreading of *Cep78*$^{+/-}$ and *Cep78*$^{-/-}$ mice testicular tissues to detect ratios of spermatocytes at leptotene, zygotene, pachytene, and diplotene stages. Our results show *Cep78* loss does not affect spermatocyte development at all four stages (*Figure 5O–P*), implying that *Cep78* deletion exerted no effect on the prophase of meiosis I.

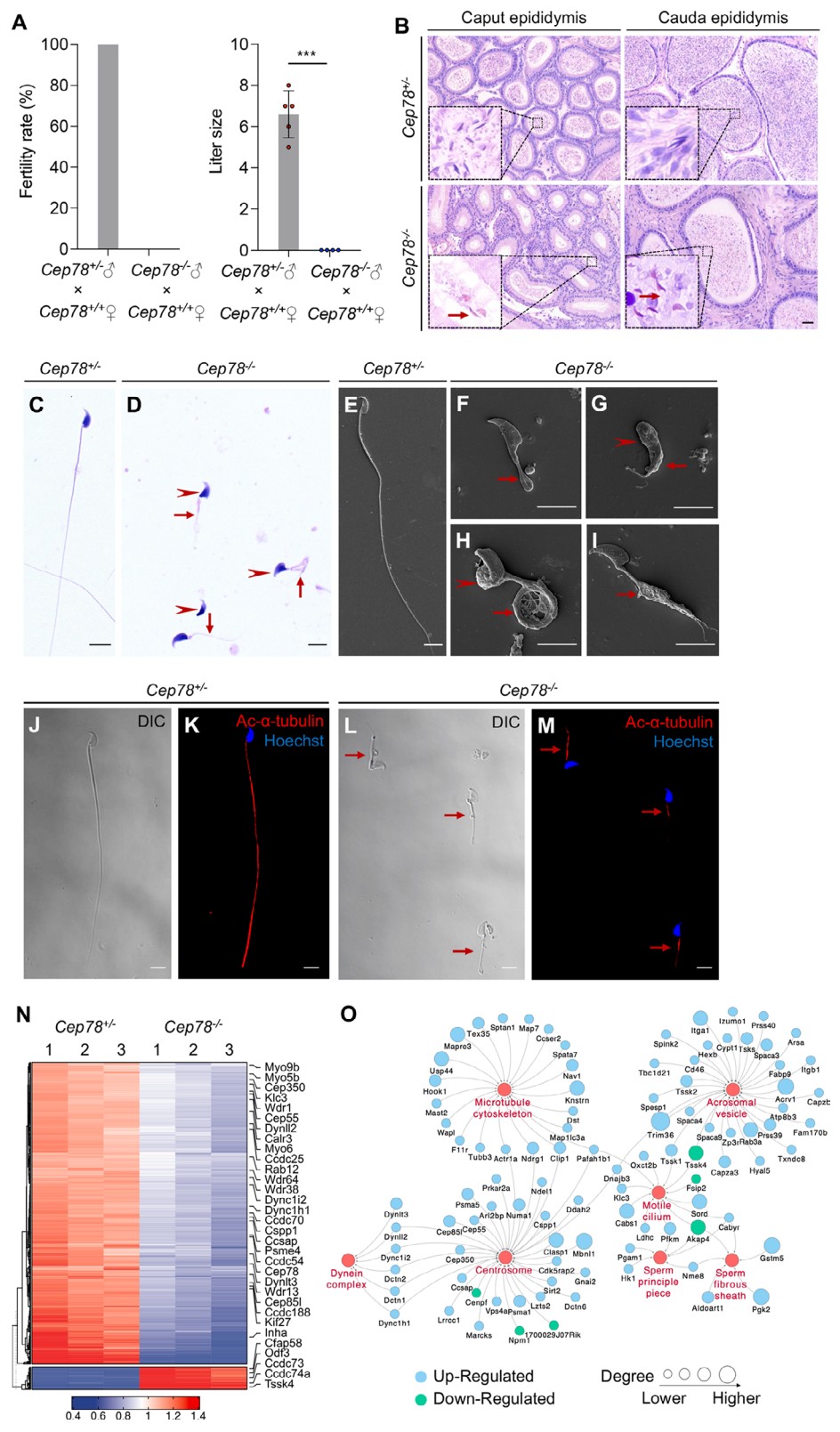

**Figure 3.** *Cep78⁻/⁻* mice present male infertility and morphological abnormalities of sperm. (**A**) Fertility rates (left) and litter sizes (right) of *Cep78⁺/⁻* and *Cep78⁻/⁻* male mice crossed with *Cep78⁺/⁺* female mice, n=4 for *Cep78⁺/⁻* male mice crossed with *Cep78⁺/⁺* female mice, n=5 for *Cep78⁺/⁻* male mice crossed with *Cep78⁺/⁺* female mice, two-tailed Student's t-test. ***, p<0.001. (**B**) Hematoxylin-eosin (H&E) staining was utilized to observe the

*Figure 3 continued on next page*

*Figure 3 continued*

structures of caput and cauda epididymis in *Cep78*[+/−] and *Cep78*[+/−] mice. Scale bar, 100 μm. (**C–D**) Spermatozoa from *Cep78*[+/−] (**C**) and *Cep78*[−/−] (**D**) mice were observed with H&E staining. Abnormal heads and short, missing, or coiled flagella were indicated by arrow heads and arrows respectively. Scale bar, 5 μm. (**E–I**) Spermatozoa of *Cep78*[+/−] (**E**) and *Cep78*[−/−] (**F–I**) mice were observed by scanning-electron microscopy (SEM). Abnormal heads were represented by arrow heads. Absent, short, coiled, or multi flagella were indicated by arrows. (**J–M**) Spermatozoa of *Cep78*[+/−] and *Cep78*[−/−] mice were pictured using differential interference contrast (**J and L**) and were stained with anti-Ac-α-tubulin (red) and Hoechst (blue) to visualize axonemes and heads, respectively (**K and M**). (**N**) Heatmap showing differentially expressed proteins between *Cep78*[+/−] and *Cep78*[−/−] mice elongating spermatids. (**O**) Network analysis of gene ontology (GO) cellular components terms enriched in differential expressed proteins between *Cep78*[+/−] and *Cep78*[−/−] mice elongating spermatids. Scale bar, 5 μm.

The online version of this article includes the following source data and figure supplement(s) for figure 3:

**Source data 1.** Original numbers used for quantification of *Figure 3A*.

**Source data 2.** Hematoxylin-eosin (H&E) staining, scanning-electron microscopy (SEM), and immunofluorescence analysis images in *Figure 2B–M*.

**Figure supplement 1.** Statistics for testis weights, body weights, and testis/body weight ratios of *Cep78*[+/+], *Cep78*[+/−], and *Cep78*[−/−] mice.

**Figure supplement 1—source data 1.** Original numbers used for quantification in *Figure 3—figure supplement 1A-C*.

**Table 1.** Semen characteristics and sperm morphology in *Cep78*[−/−] male mice and the Chinese man carrying *CEP78* mutation.

| | *Cep78*[−/−] **male mice** | Cep78[+/−] **male mice** | **Patient** | **Reference value*** |
|---|---|---|---|---|
| **Semen parameters** | | | | |
| Semen volume (mL) | - | - | 2.6 | 1.5 |
| Sperm concentration ($10^6$/mL) | 9.48±1.75 | 30.45±3.21 | 14.298 | 15 |
| Motility (%) | 8.44±1.90 | 71.45±2.24 | 11.428 | 40 |
| Progressive motility (%) | 0 | 22.72±2.24 | 2.857 | 32 |
| **Sperm morphology** | | | | |
| Normal spermatozoa (%) | 0 | 84.07±1.78 | 0 | 4 |
| Abnormal head (%) | 92.10±3.68 | 3.88±0.67 | 95.42 | - |
| Abnormal neck (%) | 82.61±1.39 | 3.89±1.22 | 85.62 | - |
| Abnormal flagella (%) | 95.67±1.98 | 12.50±1.60 | 97.39 | - |
| Short flagella (%) | 39.08±2.74 | 2.15±0.76 | 22.88 | - |
| Absent flagella (%) | 32.26±2.58 | 1.73±0.40 | 21.57 | - |
| Coiled flagella (%) | 7.71±0.98 | 3.45±0.46 | 15.69 | - |
| Multi flagella (%) | 6.01±1.29 | 1.30±0.30 | 10.46 | - |
| Cytoplasm remains (%) | 4.51±1.29 | 1.94±0.65 | 15.03 | - |
| Irregular caliber (%) | 2.59±1.74 | 0.21±0.37 | 6.54 | - |
| Angulation (%) | 3.49±2.06 | 1.73±0.40 | 5.23 | - |

[1]The lower limit of reference value (WHO laboratory manual for the examination and processing of human sperm).

*The lower limit of reference value (WHO laboratory manual for the examination and processing of human sperm).

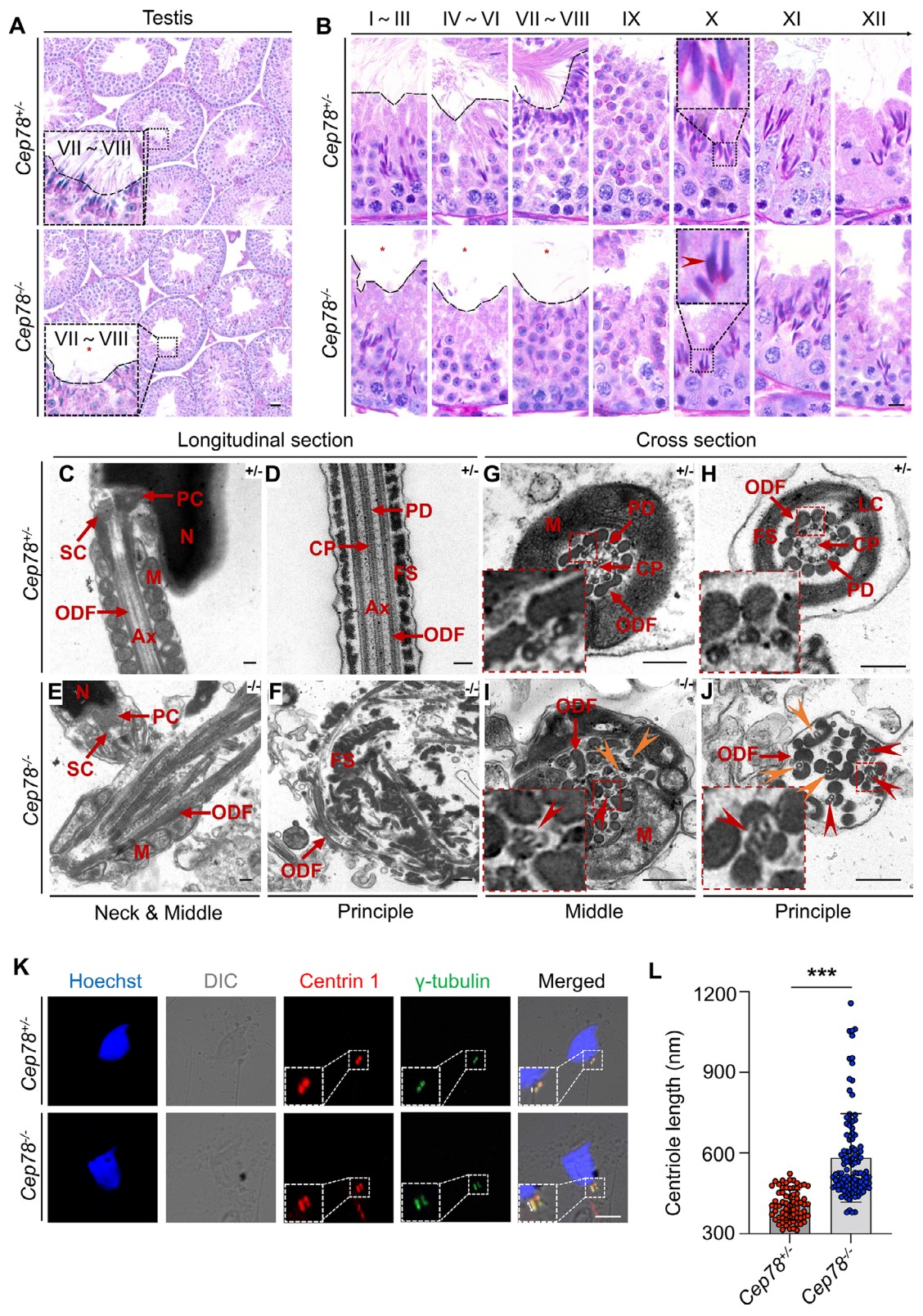

**Figure 4.** Defective microtubule arrangements and elongated centrioles in sperm flagella of *Cep78⁻/⁻* mice. (**A**) Paraffin sections of testicular tissues from *Cep78⁺/⁻* and *Cep78⁻/⁻* male mice were stained with periodic acid-Schiff (PAS). Defects of sperm tails were indicated by asterisk. Scale bar, 50 μm. (**B**) Twelve stages of spermatogenesis in *Cep78⁺/⁻* and *Cep78⁻/⁻* male mice were presented by PAS-stained paraffin sections of seminiferous tubules. Lack of sperm tails and abnormal nuclei shape of elongated spermatids were indicated by asterisk and arrowhead, respectively. Schematic diagrams

*Figure 4 continued on next page*

*Figure 4 continued*

were attached. Scale bar, 10 µm. (**C–J**) Transmission electron microscopy (TEM) was applied to visualize ultra-structures of *Cep78*$^{+/-}$ and *Cep78*$^{-/-}$ spermatozoa in longitudinal sections of neck and middle pieces (**C and E**) and principal pieces (**D and F**), and in cross sections of middle pieces (**G and I**) and principal pieces (**H and J**). Triplet and singlet microtubules were indicated by red and orange arrow heads (**I–J**), respectively. Scale bar, 200 nm. N, nucleus; M, mitochondria; Ax, axoneme; SC, segmented column of the connecting piece; PC, proximal centriole; ODF, outer dense fiber; PD, peripheral microtubule doublets; CP, central pair of microtubules; FS, fibrous sheath; LC, longitudinal column. (**K–L**) *Cep78*$^{+/-}$ and *Cep78*$^{-/-}$ mice spermatozoa were stained with anti-Centrin 1 and anti-γ-tubulin to reveal the structure of centrosome (**K**). Scale bar, 5 µm. (**L**) Length of Centrin 1 positive centriole was quantification and compared (n=70 for *Cep78*$^{+/-}$ and n=107 *Cep78*$^{-/-}$, two-tailed Student's t-test). ***, p<0.001.

The online version of this article includes the following source data for figure 4:

**Source data 1.** Original numbers used for quantification in *Figure 4L*.

**Source data 2.** Periodic acid-Schiff (PAS) staining, transmission electron microscopy (TEM), and immunofluorescence analysis images in *Figure 4A-K*.

The manchette structure, consisting of a series of parallel microtubule bundles that extend from perinuclear rings of nucleus to distal cytoplasm and are closely proximity to or parallel to nuclear membranes, regulates sperm head formation during spermiogenesis (*Kierszenbaum and Tres, 2004*). We herein revealed various abnormalities in manchette formation and acrosomal biogenesis in *Cep78*$^{-/-}$ spermatids, including an abnormal nuclear constriction at the site of perinuclear rings (*Figure 5D*), ectopic and asymmetric perinuclear rings together with disordered manchette microtubules (*Figure 5E–F– and K*), abnormal acrosome and nuclei invagination (*Figure 5L*), and detached acrosome from nuclear membrane (*Figure 5M*). The defective spermatid manchette formation in *Cep78*$^{-/-}$ mice led to peculiar shaping of nucleus and abnormal sperm head formation. Therefore, the above findings indicate that the absence of *Cep78* generates defective spermatid head formation.

## *CEP78* mutation-caused male infertility and MMAF in human patient with CRD

We have previously linked the *CEP78* c.1629–2A>G mutation with autosomal recessive CRDHL and revealed a 10 bp deletion of *CEP78* exon 14 in mRNA extracted from white blood cells of patient carrying *CEP78* c.1629–2A>G mutation (*Fu, 2017*). We further explored whether c.1629–2A>G mutation, previously reported by *Fu, 2017*, would disturb CEP78 protein expression and male fertility. Blood sample was collected from this patient and an unaffected control for protein extraction. Our immunoblotting results reveal neither full length CEP78 protein nor truncated CEP78 protein in white blood cells of patient carrying *CEP78* c.1629–2A>G mutation (*Figure 6—figure supplement 1*), suggesting that this mutation leads to loss of CEP78 protein.

Since Cep78 deprivation led to male infertility and sperm flagellar defects in mice, and *CEP78* c.1629–2A>G mutation caused degradation of CEP78 protein in human, we thus investigated reproductive phenotype of the previously reported male patient, who carried homozygous *CEP78* c.1629–2A>G mutation and was diagnosed with CRDHL (*Fu, 2017*). The patient was infertile. His semen volume was 2.6 mL, within the normal range (*Table 1*). However, CASA revealed obvious reductions in sperm concentration, motility, and progressive motility in the *CEP78*-mutated man according to the standard of the WHO guidelines (*Table 1*). In addition, impaired sperm movement of the patient is presented in *Figure 6—video 1*.

The patient's semen sample was further submitted for morphological analyses using light microscopy and SEM, both of which showed typical MMAF phenotypes, such as abnormal heads (95.42%), necks (85.62%), and flagella (97.39%; *Figure 6A–J* and *Table 1*). A great diversity of abnormal flagella was identified, short (*Figure 6B and G*), coiled (*Figure 6C, I*), absent (*Figure 6D and H*), and multiple (*Figure 6E and J*). The patient's sperm also had a series of sperm head abnormalities, such as large acrosome area (*Figure 6B*), pear shaped nucleus (*Figure 6C*), conical nucleus (*Figure 6D*), and excessive acrosome vacuoles (*Figure 6E*). TEM was further applied to visualize the ultra-structures of spermatozoa in the male patient carrying *CEP78* c.1629–2A>G mutation. Based on our data, multiple ultra-structural abnormalities were observed in the spermatozoa of case as compared to those of age-matched healthy male individual (*Figure 6K–R*). For example, longitudinal sections showed disordered arrangements of mitochondrial sheaths, severe axonemal disorganization, and fibrous sheath hyperplasia (*Figure 6L and N*). Instead of the typical '9+2' microtubule structure in exoneme of normal sperm flagella (*Figure 6O and Q*), absent or reduced central-pair microtubules and disarranged peripheral microtubule doublets were frequently observed in cross sections (*Figure 6P and R*).

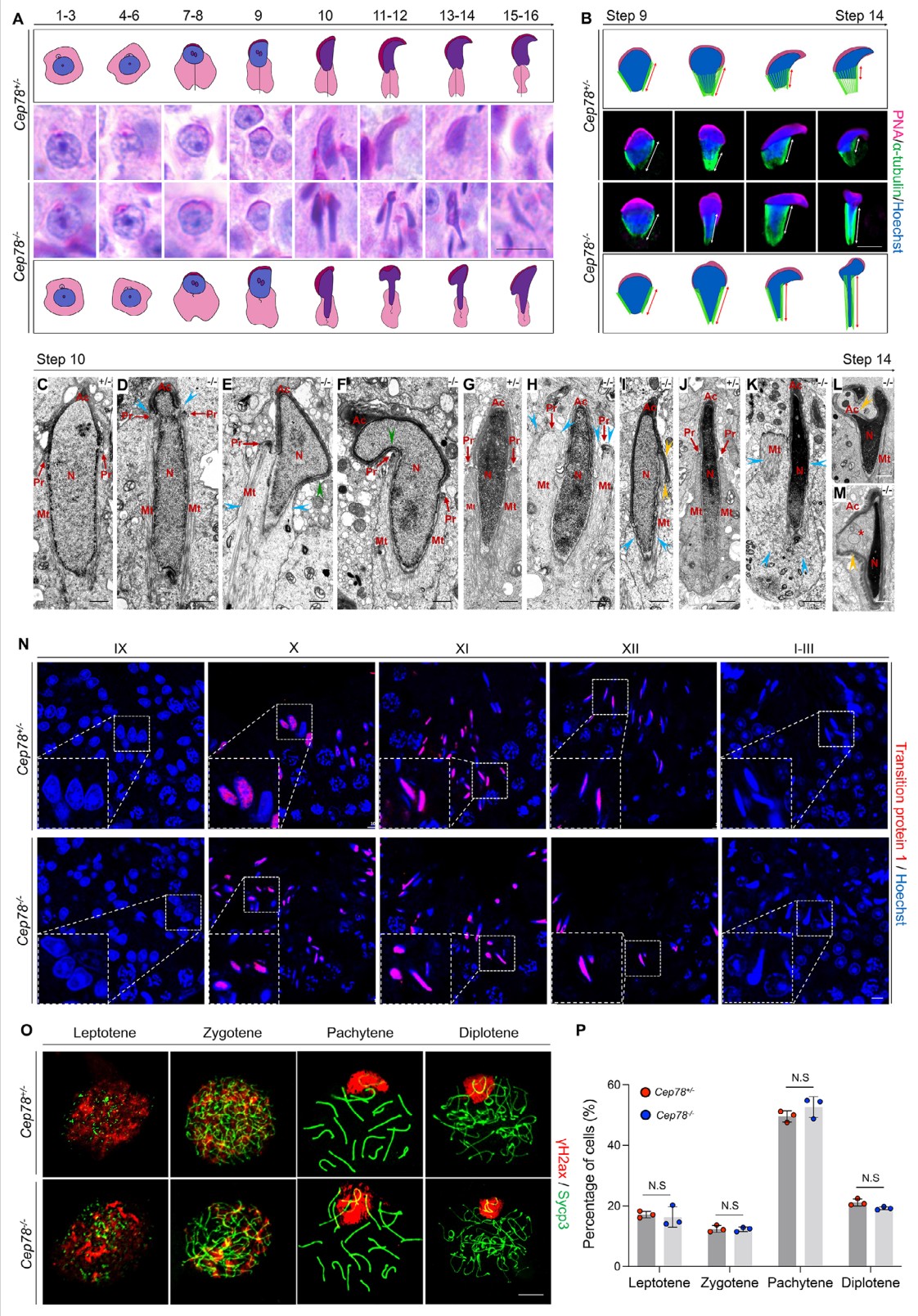

**Figure 5.** *Cep78⁻/⁻* testes present abnormalities in spermatid head formation during spermiogenesis. (**A**) Sixteen steps of spermiogenesis in *Cep78⁺/⁻* and *Cep78⁻/⁻* male mice were presented by periodic acid-Schiff (PAS)-stained paraffin sections of seminiferous tubules. Lack of sperm tails and abnormal nuclei shape of elongated spermatids were indicated by asterisk and arrowhead, respectively. Schematic diagrams were attached. Scale bar, 10 μm. (**B**) Elongated spermatids from stages 9 to 14 from *Cep78⁺/⁻* and *Cep78⁻/⁻* mice were stained with peanut agglutinin (PNA; red) and α-tubulin (green) to

*Figure 5 continued on next page*

*Figure 5 continued*

see elongation of manchette structures. Nuclei were counterstained with Hoechst (blue). Manchette structures were recognized by double-head arrows. Scale bar, 5 μm. Schematic diagrams were attached. (**C–M**) Transmission electron microscopy (TEM) was used to visualize ultra-structures of *Cep78⁺ᐟ⁻* and *Cep78⁻ᐟ⁻* spermiogenic spermatids from steps 10 to 14. Scale bar, 500 nm. N, nucleus; Ac, acrosome; Pr, perinuclear ring; Mt, manchette. Abnormal structures of manchette were indicated by celeste arrow heads (**D, E, H, I, and K**). Green arrow heads pointed at abnormal bend of spermatid heads (**E and F**). Yellow arrow heads represented abnormal acrosomes (**I, L, and M**). Asterisk indicated expanded perinuclear space (**M**). (**N**) Paraffin sections of testicular seminiferous tubules from stage IX of spermatogenesis to stage I in *Cep78⁺ᐟ⁻* and *Cep78⁻ᐟ⁻* mice were stained with anti-transition protein 1 (red) and Hoechst (blue) to observe nuclear condensation. Scale bar, 5 μm. (**O–P**) Immunofluorescence staining of anti-γH2ax (red) and anti-Sycp3 (green) in chromosome spreads of spermatocytes from the testes of *Cep78⁺ᐟ⁻* and *Cep78⁻ᐟ⁻* mice. Scale bar, 5 μm. Representative image along with the quantification results was shown (n=219 for *Cep78⁺ᐟ⁻* spermatocytes, n=208 for *Cep78⁻ᐟ⁻* spermatocytes, two-tailed Student's t-test, N.S, not significant).

The online version of this article includes the following source data for figure 5:

**Source data 1.** Original numbers used for quantification in *Figure 5P*.

**Source data 2.** Periodic acid-Schiff (PAS) staining analysis images in *Figure 5A*.

**Source data 3.** Immunofluorescence and transmission electron microscopy (TEM) analysis images in *Figure 5B–O*.

Similar to the *Cep78⁻ᐟ⁻* mice data, triplet microtubules unique to proximal centrioles in sperms were also observed in both principal and middle pieces of patient's sperm flagella (*Figure 6P and R*). Taken together, this patient carrying *CEP78* c.1629–2A>G mutation presented typical MMAF phenotypes.

## CEP78 interacted physically with IFT20 and TTC21A and is essential for their interaction and stability to regulate centriole and cilia lengths

To reveal the interaction between CEP78 and other proteins during spermiogenesis, we performed anti-Cep78 immunoprecipitation (IP) coupled with quantitative MS (IP-MS) on testicular lysates of *Cep78⁺ᐟ⁻* and *Cep78⁻ᐟ⁻* mice. A total of 16 ciliary proteins were initially screened out by proteomic analyses (*Table 2*), among which Ift20 and Ttc21A are essential for sperm flagella assembly and male fertility. These two proteins bound together as a dimer, and loss of IFT20 or TTC21A was found to cause male infertility and MMAF phenotypes (*Liu et al., 2019*; *Zhang et al., 2016a*), and thus, we hypothesized that CEP78 regulates human and mice phenotypes by interacting with IFT20 and TTC21A. To confirm this hypothesis, we first tested whether the three proteins bind with each other using co-immunoprecipitation (co-IP) and immunoblotting analyses in HEK293T cells overexpressing tagged proteins (*Figure 7—figure supplement 1*) and with Ap80-NB (*Yi et al., 2013*) as a relevant negative control (*Figure 7A–C*, *Figure 7—figure supplement 2A–C*). Protein interactions between CEP78 and IFT20 (*Figure 7A*, *Figure 7—figure supplement 2A*), CEP78 and TTC21A (*Figure 7B*, *Figure 7—figure supplement 2B*), and IFT20 and TTC21A (*Figure 7C*, *Figure 7—figure supplement 2C*) were identified, supporting our hypothesis.

We next detected whether Cep78 depletion would cause instability of the protein interaction between Ift20 and Ttc21a, and disturb its functions. As revealed by co-IP and immunoblotting results, the binding between Ift20 and Ttc21a was severely disrupted upon Cep78 knockout (*Figure 7D*), indicating that Cep78 is essential for the interaction between Ift20 and Ttc21a. Meanwhile, the size exclusion chromatography result demonstrated that endogenous testicular Cep78, Ift20, and Ttc21a proteins were co-fractionated in a complex ranging in size of 158–670 kDa (*Figure 7E*). Cep78 was previously reported to localize at the distal half of the centriole (*Hossain, 2017*; *Gonçalves et al., 2021*), our 3D structured illumination microscopy (SIM) showed that Ift20 and Ttc21a localized peripheral to Cep78 at the top view of the centriole, and Ift20 and Ttc21a co-localized at one end of Cep78-positive centriole part (*Figure 7F*).

In addition, we found that expression of both Ttc21a and Ift20 decreased in testes of *Cep78⁻ᐟ⁻* mice compared to *Cep78⁺ᐟ⁻* mice (*Figure 7G–I*). Similarly, decreased Ttc21a and Ift20 protein levels were observed in *Cep78* siRNA transfected NIH3T3 cells compared with cells transfected with scramble siRNA (*Figure 7J–M* and *Figure 7—figure supplement 1*), which could be rescued by expression of exogenous Cep78-HA with synonymous mutations at siRNA binding region (Cep78ˢʸⁿ-HA; *Figure 7J–M*, *Figure 7—figure supplement 3*, Table 4). These results show that disruption of the Cep78 potentially affects stability of its interacting proteins, Ift20 and Ttc21a.

We then aimed to determine the functions of Ift20 and Ttc21a. Cep78 is a centrosomal protein that was reported to be located at distal end of centrioles. 3D-SIM results showed that Ift20 and Ttc21a localized at one end of Cep78-positive centriole part (*Figure 7F*). And we found that all three

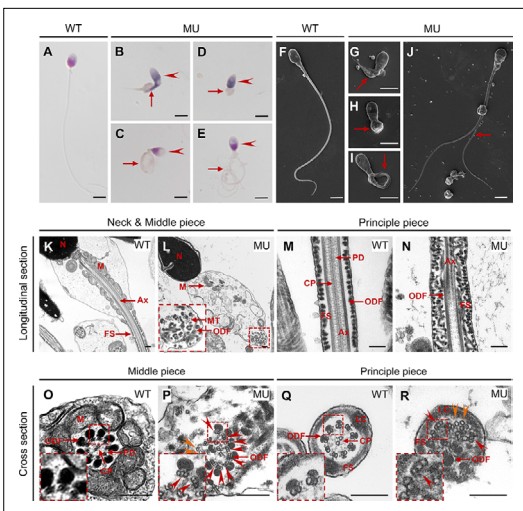

**Figure 6.** *CEP78* mutation causes multiple morphological abnormalities of sperm flagella (MMAF) phenotypes in human. (**A–E**) Hematoxylin-eosin (H&E) staining was utilized to see structures of spermatozoa from a healthy control (**A**) and a patient carrying homozygous *CEP78* c.1629–2A>G mutation (**B–E**). Scale bar, 5 μm. Arrow heads represented abnormal sperm heads. Short, coiled, absent, or multi flagella were indicated by arrows. (**F–J**) Ultra-structures of spermatozoa from the healthy control (**F**) and the patient with *CEP78* mutation (**G–J**) were observed using scanning-electron microscopy (SEM). Short, coiled, absent or multi flagella, and cytoplasm remains were indicated by arrows. Scale bar, 5 μm. (**K–R**) Transmission electron microscopy (TEM) was applied to visualize ultra-structures of spermatozoa from the healthy control and the patient in longitudinal sections of neck and middle pieces (**K–L**) and principal pieces (**M–N**), and in cross sections of middle pieces (**O–P**) and principal pieces (**Q–R**). Triplet and singlet microtubules were indicated by red and orange arrow heads (**P and R**), respectively. Scale bar, 200 nm (**K–R**). N, nucleus; M, mitochondria; MT, microtubules; Ax, axoneme; FS, fibrous sheath; ODF, outer dense fiber; PD, peripheral microtubule doublets; CP, central pair of microtubules; LC, longitudinal column.

The online version of this article includes the following video, source data, and figure supplement(s) for figure 6:

**Source data 1.** Hematoxylin-eosin (H&E) staining, scanning-electron microscopy (SEM), and transmission electron microscopy (TEM) analysis images in *Figure 6A–R*.

**Figure supplement 1.** *CEP78* mutation causes absence of CEP78 protein.

**Figure supplement 1—source data 1.** Uncropped blots of *Figure 6—figure supplement 1*.

**Figure 6—video 1.** Sperm motility analysis video of the patient carrying *CEP78* c.1629–2A>G mutation. https://elifesciences.org/articles/76157/figures#fig6video1

proteins were located at the base of the cilium (*Figure 7—figure supplement 4*), indicating their roles in ciliogenesis. To respectively clarify the cilliogenesis function of Cep78, Ift20, and Ttc21a, we investigated the functional consequences of their knockdown (*Figure 7N and O*). Efficiencies for *Cep78*, *Ift20*, and *Ttc21a* knockdown using *Cep78*-siRNA, *Ift20*-siRNA, and *Ttc21a*-siRNA and rescue by Cep78syn-HA, Ift20syn-Flag, and Ttc21asyn-Flag with synonymous mutations at their siRNA targeting sequence were confirmed by quantitative PCR (Q-PCR; *Figure 7—figure supplement 3A–C*) and western blotting (*Figure 7J–M* and *Figure 7—figure supplement 3D, E, H, I*). With ciliary axonemes and centrioles labeled with anti-Ac-α-tubulin and anti-γ-tubulin, respectively, we found that cilium length was reduced after *Cep78*, *Ift20*, or *Ttc21a* knockdown (*Figure 7N and O*) and could be rescued by exogenous expression of Cep78syn-HA, Ift20syn-Flag, and Ttc21asyn-Flag (*Figure 7N and O*).

Moreover, since we observed prolonged centriole (*Figure 4K*) and triplet flagella microtubules (*Figure 4I, J and K* and *Figure 6P and R*) in *Cep78*−/− mouse spermatids and the *CEP78* mutant patient, we examined the effects of *Cep78*, *Ift20*, and *Ttc21a* knockdown on the regulation of centriole length (*Figure 7P and Q*). We found that the centrioles were longer in NIH3T3 cells with *Cep78* or *Ift20* knockdown, and the length changes were reverted by overexpression of Cep78syn-HA and Ift20syn-Flag (*Figure 7P and Q*). However, we observed no change in centriole length after *Ttc21a* knockdown or rescue (*Figure 7P and Q*). Additionally, we also observed that the centriolar localization of Ift20 protein was disrupted after Cep78 knockdown (*Figure 7R and S*), and depletion of Cep78 downregulated Ift20 and Ttc21a (*Figure 7G, H, J and M*), while knockdown of *Ift20* or *Ttc21a* did not significantly affect the protein level of Cep78 and other interacting proteins (*Figure 7—figure supplement 3D, F, G, H, J, K*), indicating that Cep78 maintains centriole homeostasis and cilliogenesis by recruiting Ift20 and Ttc21a.

Thus, our data suggest that insufficiency of Cep78 and its interacting proteins caused cilia shortening and centriole elongation, which is consistent with phenotypes we observed in *Cep78*−/− mice. Collectively, our findings demonstrate that CEP78 interacts with IFT20 and TTC21A to regulate axonemal and centrosomal functions.

**Table 2.** Cep78[*] interacting proteins identified with immunoprecipitation coupled with quantitative mass spectrometry (IP-MS).

| Genes | Proteins | Q-value | Fold change (Cep78[+/−] vs Cep78[−/−]) |
|---|---|---|---|
| Ehd1 | EH domain-containing protein 1 | 0 | 3.69472331 |
| Traf3ip1 | TRAF3-interacting protein 1 | 0 | 4.46937703 |
| Armc4 | Armadillo repeat-containing protein 4 | 0 | 6.50427256 |
| Cep131 | Centrosomal protein of 131 kDa | 0 | 10.1830839 |
| Cct8 | T-complex protein 1 subunit theta | 0 | 27.7594376 |
| Dync2h1 | Cytoplasmic dynein 2 heavy chain 1 | 0 | 3.52034955 |
| Dync2li1 | Cytoplasmic dynein 2 light intermediate chain 1 | 0 | 2.35693042 |
| Hspb11 | Intraflagellar transport protein 25 homolog | 0 | 2.591138 |
| Ift122 | Intraflagellar transport protein 122 homolog | 0 | 73,932 |
| Ift140 | Intraflagellar transport protein 140 homolog | 0 | 20,456 |
| Ift20 | Intraflagellar transport protein 20 homolog | 0 | 16.4424842 |
| Pfkm | ATP-dependent 6-phosphofructokinase, muscle type | 0 | 90,119 |
| Spef1 | Sperm flagellar protein 1 | 0 | 3.34178683 |
| Ttc21a | Tetratricopeptide repeat protein 21 A | 0 | 3.12239761 |
| Ttc21b | Tetratricopeptide repeat protein 21B | 0 | 3.17696419 |
| Ttc30a1 | Tetratricopeptide repeat protein 30A1 | 0.0024355 | 38,700 |

[*]The antibody against mouse Cep78 (p457-741) was used for IP-MS analysis.

## Discussion

The major findings of our study are as follows: we found *CEP78* as the causal gene of CRD with male infertility and MMAF using *Cep78*[−/−] mice. A male patient carrying *CEP78* c.1629–2A>G mutation, whom we previously reported to have CRD (*Fu, 2017*), was found to have male infertility and MMAF in this study. Cep78 interacted with sperm flagella formation enssential proteins IFT20 and TTC21A (*Figure 8*), which are essential for sperm flagella formation (*Liu et al., 2019*; *Zhang et al., 2016a*). Cep78 played an important role in the interaction and stability of these proteins, which regulate flagella formation and centriole length in spermiogenesis.

CEP78, which belongs to the centrosomal and ciliary protein family, mutations in other centrosomal genes, such as *CEP19*, *CEP164*, *CEP250*, and *CEP290*, could disrupt ciliary assembly and generate retinopathy (*Kubota et al., 2018*; *Yıldız Bölükbaşı et al., 2018*; *Chaki et al., 2012*; *Valente et al., 2006*; *Baala et al., 2007*; *den Hollander et al., 2006*). However, there is no evidence that loss of the above centrosomal proteins can simultaneously cause CRD and male infertility.

*Ascari et al., 2020* reported three patients with male reproductive phenotypes, among whom only one patient had mutations only in *CEP78* gene and absence of its full-length protein expression. The patient had compound heterozygous mutations with a missense and a nonsense mutation (c.[449T>C];[1462–1G>T] p.[Leu150Ser];[?]) and exhibited asthenoteratozoospermia and male infertility, with the sperm phenotype less severe than that observed in our study. The patient in our study showed phenotype of oligoasthenoteratozoospermia with MMAF, consistent with the phenotype observed in Cep78 knockout mice. Whether the patient reported by *Ascari et al., 2020* had truncated CEP78 by nonsense mutation was not analyzed, which might contribute the phenotypic differences between patients.

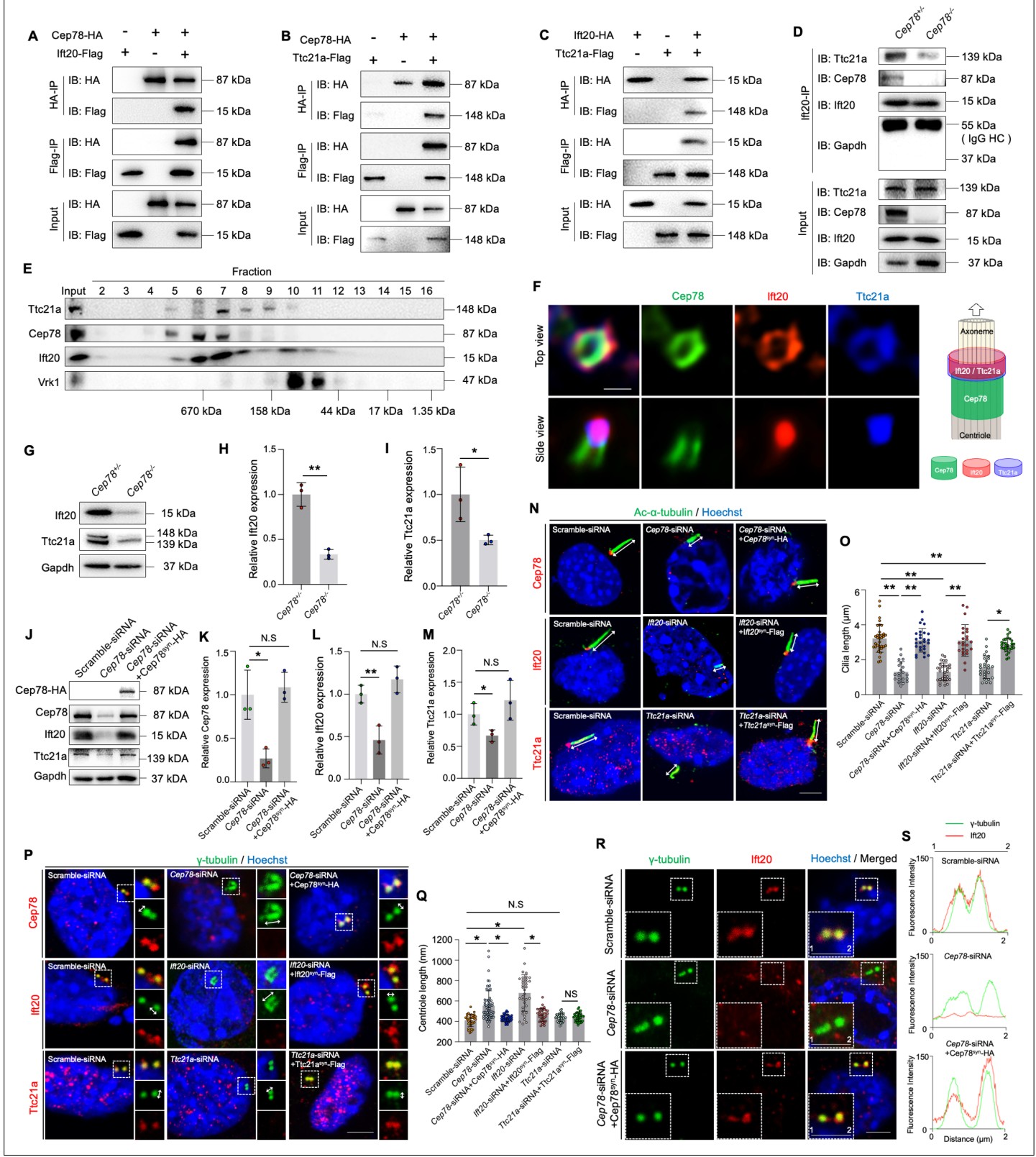

**Figure 7.** CEP78 interacted physically with IFT20 and TTC21A to regulate ciliary axonemal and centriole elongations. (**A–C**) HEK293T cells were co-transfected with plasmids expressing Cep78-HA and Ift20-Flag (**A**), Cep78-HA and Ttc21a-Flag (**B**), or Ift20-HA and Ttc21a-Flag (**C**). Cellular lysates were immunoprecipitated with an anti-HA or anti-Flag antibody in 1% Sodium dodecyl sulfate (SDS) and then immunoblotted (IB) with anti-HA or anti-Flag antibodies. (**D**) Lysates of testicular tissues from *Cep78*[+/–] and *Cep78*[–/–] male mice were immunoprecipitated with anti-Ift20 antibody in 1% SDS and

*Figure 7 continued on next page*

*Figure 7 continued*

then immunoblotted with anti-Ttc21a, anti-Cep78, anti-Ift20, and anti-Gapdh antibodies. To balance the protein amount of Ift20 and Ttc21a between *Cep78*[+/−] and *Cep78*[−/−] testes, the amount of input and proteins for immunoprecipitation in *Cep78*[−/−] was three times that of *Cep78*[+/−]. HC, Heavy chain. (**E**) Mouse testicular lysates were chromatographed on a Superpose-6 size exclusion column, and the resulting fractions (Fractions 2–16) were western blotted with the indicated antibodies of Ttc21a, Cep78, and Ift20 with Vaccinia related kinase 1 (Vrk1) as a negative control. (**F**) 3D structured illumination microscopy (SIM) analysis (left) and schematic diagram (right) of localizations of CEP78, Ift20, and Ttc21a in NIH3T3 cells, scale bar = 200 nm. (**G–I**) Immunoblotting showed Ift20 and Ttc21a expressions in lysates of testes from *Cep78*[+/−] and *Cep78*[−/−] mice. (**G**) Representative images along with the quantification results were shown (**H–I**); three biological replications for *Cep78*[+/−] and *Cep78*[−/−] testes, two-tailed Student's t-test.(**J–M**) Expression levels of Cep78 (**J and K**), Ift20 (**J and L**), Ttc21a (**J and M**), and Cep78-HA (**J**) proteins were tested in NIH3T3 cells transfected with scramble siRNA, *Cep78*-siRNA, and *Cep78*-siRNA+Cep78[syn]-HA through western blot (three biological replications for scramble siRNA, *Ift20*-siRNA, and *Ift20*-siRNA+Ift20[syn]-Flag, two-tailed Student's t-test). (**N–Q**) Cilia (**N**) and centrioles (**P**) in cells transfected with scramble siRNA, *Cep78*-siRNA, *Ift20*-siRNA, *Ttc21a*-siRNA, *Cep78*-siRNA+Cep78[syn]-HA, *Ift20*-siRNA+Ift20[syn]-Flag, and *Ttc21a*-siRNA+Ttc21a[syn]-Flag were stained with anti-Ac-α-tubulin and anti-γ-tubulin, respectively. Nuclei were counterstained with Hoechst (blue). Cilium and centriole structures were recognized by double-head arrows. Scale bar, 2 μm. The quantification results were shown. (**O and Q**) Comparisons in cilia and centriole size between the above-mentioned samples were addressed by accumulated data from three independent experiments, for cilia length: n=34 for scramble-siRNA, n=21 for *Cep78* siRNA, n=27 for *Cep78*-siRNA+Cep78[syn]-HA, n=30 for *Ift20* siRNA, n=24 for *Ift20*-siRNA+Ift20[syn]-Flag, n=27 for *Ttc21a* siRNA, n=30 for *Ttc21a*-siRNA+Ttc21a[syn]-Flag; for centriole length: n=34 for scramble-siRNA, n=67 for *Cep78* siRNA, n=39 for *Cep78*-siRNA+Cep78[syn]-HA, n=35 for *Ift20* siRNA, n=36 for *Ift20*-siRNA+Ift20[syn]-Flag, n=33 for *Ttc21a* siRNA, n=42 for *Ttc21a*-siRNA+Ttc21a[syn]-Flag; one-way ANOVA, with Dunnet's multiple comparison test amongst all groups. (**R and S**) Cells transfected with scramble siRNA, *Cep78*-siRNA, and *Cep78*-siRNA+Cep78[syn]-HA were stained with anti-Ift20 and anti-γ-tubulin to reveal the intensities and locations of Ift20 and centrioles. Fluorescence intensity traces are plotted. Scale bar, 2 μm. N.S, not significant, *, p<0.05; **, p<0.01.

The online version of this article includes the following source data and figure supplement(s) for figure 7:

**Source data 1.** Uncropped blots of *Figure 7A–E, G and J*.

**Source data 2.** Original numbers used for quantification in *Figure 7H, I, K-M, O, Q, and S*.

**Source data 3.** Immunofluorescence analysis images in *Figure 7F, N and P* (Row 3), and R.

**Source data 4.** Immunofluorescence analysis images in *Figure 7P* (Rows 1–2).

**Figure supplement 1.** PCR test of mycoplasma in cell lines used in this study.

**Figure supplement 1—source data 1.** Uncropped gel of *Figure 7—figure supplement 1*.

**Figure supplement 2.** Co-immunoprecipitation (Co-IP) between Cep78, Ift20, and Ttc21a with negative control.

**Figure supplement 2—source data 1.** Uncropped blots of *Figure 7—figure supplement 2*.

**Figure supplement 3.** Expressions of Cep78, Ift20, and Ttc21a mRNAs in NIH3T3 cells transfected with siRNAs and rescue plasmids.

**Figure supplement 3—source data 1.** Uncropped blots of *Figure 7—figure supplement 3D, H*.

**Figure supplement 3—source data 2.** Original numbers used for quantification in *Figure 7—figure supplement 3A-C, E-G, and I-K*.

**Figure supplement 4.** Cep78, Ift20, and Ttc21a located in the base of the cilium.

**Figure supplement 4—source data 1.** Immunofluorescence analysis images in *Figure 7—figure supplement 4*.

**Figure supplement 5.** Immunoblotting showed Cp110 expressions in lysates of testes from *Cep78*[+/−] and *Cep78*[−/−] mice.

**Figure supplement 5—source data 1.** Uncropped blots of *Figure 7—figure supplement 5*.

**Figure supplement 5—source data 2.** Original numbers used for quantification in *Figure 7—figure supplement 5*.

While this study was under review, another lab in parallel reported similar findings that loss of Cep78 caused CRD, MMAF, and male infertility (*Zhang et al., 2022*). Zhang et al. reported that CEP78 regulated USP16 expression and further stabilizes Tektins via the deubiquitination pathway. Our study found that Cep78 regulated the localization, interaction, and expression levels of Ift20 and Ttc21. Defect of Ift20 or Ttc21 could cause male infertility and MMAF (*Liu et al., 2019*; *Zhang et al., 2016a*), which also observed in Cep78 knockout mice and the patient with Cep78 mutation.

In *Cep78*[−/−] photoreceptors, we observed shortened connecting cilia, spreading of microtubule doublets in the proximal half, and aberrant organizations of OS disks, which is similar to previous findings in mice with disrupted Fam161a protein (*Karlstetter et al., 2014*). Our data provide in vivo evidence for CEP78's role in maintaining structures and functions of microtubule doublets in the connecting cilium of photoreceptors.

In addition to abnormal arrangement of sperm flagella microtubule, we also found that CEP78 disruption could cause abnormal formation of manchette, a transient structure surrounding the elongating spermatid head, only present during spermatid elongation (*Kierszenbaum and Tres, 2004*; *Lehti and Sironen, 2016*). During spermiogenesis, manchette and sperm tail axoneme are

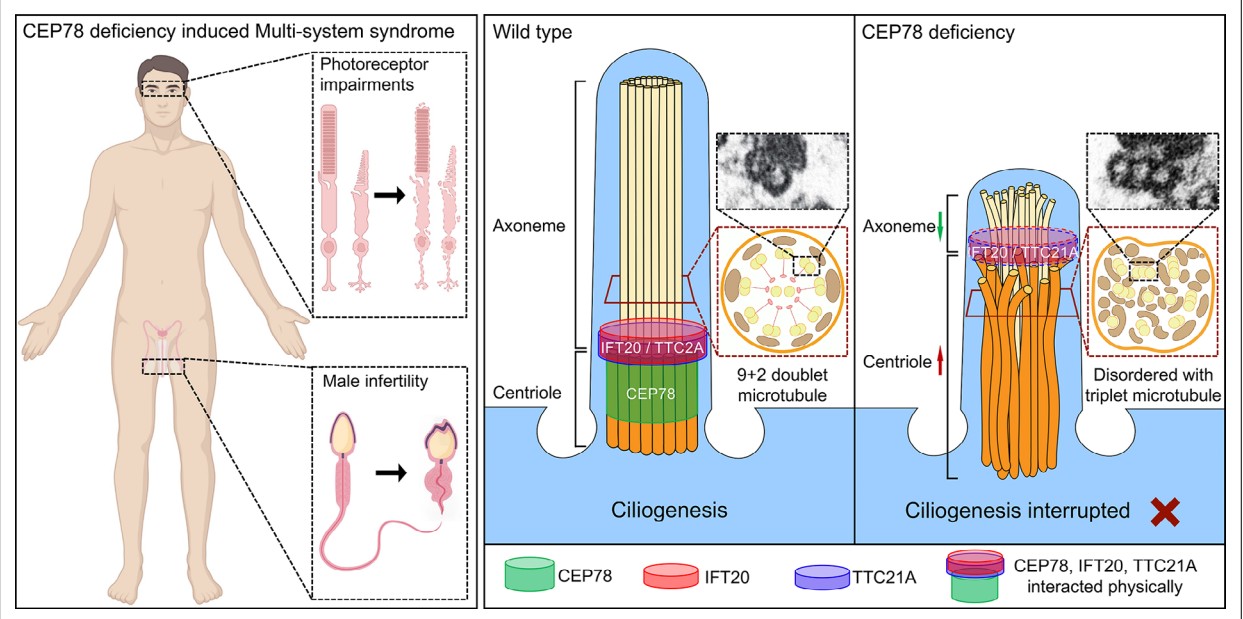

**Figure 8.** Schematic diagram of pathogenesis induced by CEP78 deficiency. Loss of CEP78 deregulated the expression of and interactions between IFT20 and TTC21A, two proteins interacted physically with CEP78. CEP78 deficiency causes elongated centrioles, shortened axonemes, abnormal presence of triplet microtubules, and disordered 9+2 microtubule axoneme.

two microtubular-based platforms that deliver cargo proteins to sperm heads and tails, respectively (*Lehti and Sironen, 2016*). However, how manchette is formed is still not fully elucidated. Most views hold that manchette microtubules are nucleated at spermatid centrosomes (*Lehti and Sironen, 2016*; *Kierszenbaum et al., 2002*), which start from nucleator γ-tubulin of the centrosomal adjunct and end at perinuclear rings (*Akhmanova et al., 2005*; *Kierszenbaum et al., 2011*). According to our data, besides disorganized manchette structures, elongated centrioles are also observed in *Cep78⁻/⁻* mice, indicating CEP78's potential role in organizing manchette microtubules. Taken together, CEP78 dysfunction specifically impairs organization of microtubular-based structures.

One previous study indicated the involvement of CEP78 in regulation of centriole length. We herein also observed stretched centriole length in *Cep78⁻/⁻* spermatids, and downregulation of the interacting protein of CEP78, Ift20. CEP78 can recruit the centriole localization of Ift20, whose knockdown causes centriole elongation. Moreover, triplet microtubules, which are specific to proximal centriole in sperm (*Avidor-Reiss and Fishman, 2019*), are aberrantly found in principal and middle pieces of sperm flagella from both the male patient with *CEP78* mutation and *Cep78⁻/⁻* mice, possibly caused by abnormal elongation of the triplet microtubules of centriole. However, such anomaly in both human and mouse sperms has never been reported before.

We found that CEP78 loss led to shortening of sperm flagellum, with Cp110 downregulated in *Cep78⁻/⁻* testis (*Figure 7—figure supplement 5*), which is inconsistent with *Gonçalves et al., 2021*'s findings that cells lacking CEP78 display upregulated CP110 and reduced cilia frequency with long cilia phenotype (*Gonçalves et al., 2021*). It seems that Cep78 may regulate formation of sperm flagellum by a different mechanism. Our analysis of interacting proteins of Cep78 showed that CEP78 interacted with two other proteins required for sperm flagellar formation, IFT20 and TTC21A. Cep78 regulates the interaction between Ift20 and Ttc21a, and loss of Cep78 led to downregulation of both Ift20 and Ttc21a. Insufficiency of Cep78 and any interacting proteins disrupts its structure and affects stability of all components, leading to cilia shortening. IFT20, belonging to the IFT family, is essential for the assembly and maintenance of cilia and flagella (*Zhang et al., 2016b*). It is required for opsin trafficking and photoreceptor OS development (*Keady et al., 2011*) and plays important roles in spermatogenesis and particularly spermiogenesis by transferring cargo proteins for sperm flagella formation (*Zhang et al., 2016b*). During spermiogenesis, IFT20 is transported along the manchette microtubules (*Kazarian et al., 2018*). IFT20 is essential to male fertility, spermatogenesis, and particularly spermiogenesis by transferring cargo proteins for sperm flagella formation. Multiple sperm

flagella abnormalities are observed in *Ift20* mutant mice, including short and kinked tails (*Zhang et al., 2016a*). TTC21A/IFT139 is also essential for sperm flagellar formation (*Liu et al., 2019*). TTC21A interacts with IFT20, and mutations in *TTC21A* gene are reported to cause MMAF in both human and mice (*Liu et al., 2019*). However, more studies are still warranted to better elucidate the biological mechanism of CEP78 in regulating features of spermatozoa and photoreceptors.

CEP78 protein is highly conserved in vertebrate organisms with human and mouse CEP78 proteins sharing over 80% similarity. Disease phenotypes and changes of ultra-structures were highly consistent between human and mice upon CEP78 depletion. Our study also reveals *Cep78*$^{-/-}$ mice model as an applicable animal model for therapeutic research of such a distinct syndrome.

In conclusion, we identify CEP78 as a causative gene for a type of syndrome involving CRD and male infertility. CEP78 dysfunction induces similar phenotypes and ultra-structure changes in human and mice, including disturbed ciliary structure in photoreceptors, defective sperm flagella structures, and aberrant spermatid head formation. We also found that CEP78 interacted physically with IFT proteins IFT20 and TTC21A and regulated their interaction and stability. The interaction between CEP78, IFT20, and TTC21A is important for the regulation of centriole and cilia length. Our study explains the etiology and animal model of the syndrome of CRD and male infertility, provides basis for molecular diagnosis, and serves as a potential target for future gene therapy.

## Materials and methods
### Ethical statement
Our study, conformed to the Declaration of Helsinki, was prospectively reviewed and approved by the ethics committee of People's Hospital of Ningxia Hui Autonomous Region ([2016] Ethic Review [Scientific Research] NO. 018) and Nanjing Medical University (NMU Ethic Review NO. [2019] 916). Signed informed consents were obtained from all individuals in the study.

### Mouse breeding
All mice were raised in a specific-pathogen-free animal facility accredited by Association for Assessment and Accreditation of Laboratory Animal Care (AAALAC) in Model Animal Research Center, Nanjing University, China. The facility provided ultraviolet sterilization, a 12 hr light/dark cycle, ad libitum access to water, and standard mouse chow diet. Mice experiments were performed in accordance with approval of the Institutional Animal Care and Use Committee of Nanjing Medical University (IACUC-1707017–8) and with the ARVO Statement for the Use of Animals in Ophthalmic and Vision Research.

### Construction of *Cep78* knockout mice using CRISPR/Cas9
CRISPR/Cas9 technology was applied to generate *Cep78* frameshift mutations in mice using the non-homolog recombination method. sgRNA were designed against exons 2–11 of *Cep78*. Cas9 mRNA and sgRNAs were synthesized by in vitro transcription assay and microinjected into the cytoplasm of single-cell C57BL/6J zygotes. The injected embryos were then transferred into oviducts of pseudo-pregnant female mice. Generated founder mice and their progenies were genotyped by sequence

**Table 3.** Primers used in this study.

| Gene/RNA | Forward primer (5′→3′) | Reverse primer (5′→3′) |
|---|---|---|
| **Genotyping** | | |
| *Cep78*-null allele | CTCTGGTCATCCCTTTGTCTAATT | TAAGACCAAACGACTTCCTCAAAC |
| Wild type *Cep78* allele | CTCTGGTCATCCCTTTGTCTAATT | CCTCTAAGAGCCTTACATAACTGG |
| **Quantitative PCR (Q-PCR)** | | |
| *Cep78* | GTTGGTTCTGAAAACTGGAATAGG | GTCCATTGGCAAATATTAACTTGAA |
| *Ift20* | TGAACTGAACAAGCTCCGAGT | GCAAGTTCCGAGCACCAAT |
| *Ttc21a* | GGCAGGATTGGGATCAGACG | GGTTTGGATTGTGGGGCTCT |
| *18S rRNA* | CATTCGAACGTCTGCCCTATC | CCTGCTGCCTTCCTTGGA |

**Table 4.** Sequences of siRNAs and synonymous mutation sequences in rescue plasmid used in this study.

| | Sequence (5'→3') | Synonymous mutation sequence in rescue plasmid (5'→3') |
|---|---|---|
| Scramble-siRNA | sense: UUCUCCGAACGUGUCACGUTT | |
| *Cep78*-siRNA | sense: GCACUUGUCUCUUGCAAAUTT | ACATTTATCCCTCCGAAC |
| *Ift20*-siRNA | sense: GUGGUCUAAUUGAGCUUGUTT | GCGGCCTGATCGAACTCGT |
| Ttc21a-siRNA | sense: GCGCCCUGAAAUCCUAUAATT | GTGCACTTAAGTCTTACAA |

analyses of the genomic DNA isolated by mice ear clipping to screen for the frameshift mutation in *Cep78*. Genotyping was performed using primer pairs for *Cep78*-null allele with an expected produce size of ~400 bp and wild type *Cep78* allele with an expected produce size of 517 bp. Primer sequences were listed in *Table 3*. Identified founder mice were crossed with wild type C57BL/6J mice to produce F1 progenies. Homozygote *Cep78*[−/−] mice were screened from F2 progenies generated by inbreeding of F1 heterozygote progenies.

## Cell culture and transfection

HEK293T and NIH3T3 cells were maintained in Dulbeccos Modified Eagle Medium (DMEM) supplemented with 10% fetal bovine serum (Invitrogen, Carlsbad, CA, USA), penicillin (100 U/mL), and streptomycin (100 g/mL) at 37°C, 5% $CO_2$. The cell lines used in this study were tested to be negative for mycoplasma with TransDetect PCR Mycoplasma detection kit (Transgen, Beijing, China; *Figure 7—figure supplement 1*). HEK293T and NIH3T3 lines were authenticated using STR profiling test by Shanghai Biowing Applied Biotechnology Co., Ltd. The open reading frame sequences of mice *Cep78*, *Ift20*, and *Ttc21a* were synthesized, amplified, and inserted into the pcDNA3.1 plasmid (GenScript, Nanjing, China) with HA/Flag sequences in-frame fused to produce recombinant plasmids Cep78-HA, Ift20-Flag, Ift20-HA, and Ttc21a-Flag, respectively. Transfection assay was conducted using Exfect transfection reagent (Vazyme, Nanjing, China).

## siRNA knockdown and rescue

Scramble-siRNA, *Cep78*-siRNA, *Ift20*-siRNA, and *Ttc21a*-siRNA were purchased from GenePharma (GenePharma, Shanghai, China) with their sequences listed in *Table 4*. Cep78[syn]-HA, Ift20[syn]-Flag, and Ttc21a[syn]-Flag were designed to have synonymous mutations in the siRNA-targeting sequence to avoid siRNA knockdown (*Table 4*). siRNAs and rescue plasmids were transfected with RNAi-mate (GenePharma, Shanghai, China) according to manufacturer's instruction.

## Immunoblotting

Immunoblotting was conducted according to a previously defined protocol (*Jiang et al., 2016*; *Liu et al., 2015*). Mice neural retina and testes were isolated for immunoblotting, respectively. Collected tissues were ground in ice-cold protein lysis buffer (Beyotime, Shanghai, China) containing protease inhibitors cocktail (Roche, Basel, Switzerland) for protein extraction. The lysates were separated on 10% sodium dodecyl sulfate-polyacrylamide gel and then transferred to a polyvinylidene fluoride membrane (Millipore, Billerica, MA, USA). Membranes were then blocked with 5% skim milk at 37°C for 1 hr, incubated with primary antibodies at 4°C overnight (*Table 5*), washed with 1×tris-buffered saline tween, and probed with corresponding horse radish peroxidase-conjugated secondary antibodies (dilution: 1:10000; ICL Inc, Newberg, Germany) at room temperature for 1 hr. Blots were developed using the ECL-western blotting system (Bio-Rad, Hercules, CA, USA).

## Electroretinogram

Mice were anesthetized intraperitoneally with a mixture of ketamine (100 mg/kg) and xylazine (10 mg/kg). Pupils were dilated with 1% cyclopentolate-HCL and 2.5% phenylephrine. After dark adaption for 8 hr, ERG was recorded under dim red light using an Espion system (Diagnosys LLC, Lowell, MA, USA) in accordance with recommendations of the International Society for Clinical Electrophysiology of Vision. ERG waves were documented in response to flashes at 0.01, 3.0, and 10.0 cd×s/m².

**Table 5.** Antibodies and dyes used in this study.

| Anti-protein | Host | Dilution and application | Supplier |
| --- | --- | --- | --- |
| CEP78-human | Rabbit | 1:1000, Immunoblotting | Abclonal (custom made) |
| GAPDH-human | Rabbit | 1:3000, Immunoblotting | Abways |
| Cep78-mouse | Rabbit | 1:1000, Immunoblotting; 1:100, Immunofluorescence | Abclonal (custom made) |
| Gapdh-mouse | Rabbit | 1:3000, Immunoblotting | Abways |
| Cone arrestin | Rabbit | 1:100, Immunofluorescence | Millipore |
| Peanut agglutinin (PNA) | | 1:500, Immunofluorescence | VectorLaboratories |
| Ac-α-tubulin | Rabbit | 1:100, Immunofluorescence (retina) | Abcam |
| Ac-α-tubulin | Mouse | 1:1000, Immunofluorescence (sperm) | Sigma-Aldrich |
| Centrin 1 | Rabbit | 1:100, Immunofluorescence | Proteintech |
| γ-tubulin | Mouse | 1:1000, Immunofluorescence | Sigma-Aldrich |
| α-tubulin | Rabbit | 1:100, Immunofluorescence | Beyotime |
| Transition protein 1 | Rabbit | 1:200, Immunofluorescence | Proteintech |
| α-tubulin | Rabbit | 1:100, Immunofluorescence | Beyotime |
| HA | Rabbit | 1:1000, Immunoblotting; 1:200, Immunofluorescence | Sigma-Aldrich |
| Flag | Rabbit | 1:1000, Immunoblotting; 1:200, Immunofluorescence | MBL |
| Ttc21a | Rabbit | 1:500, Immunoblotting, 1:100, Immunofluorescence | ABNOVA |
| Ift20 | Rabbit | 1:1000, Immunoblotting, 1:100, Immunofluorescence | Proteintech |
| Vrk1 | Mouse | 1:500, Immunoblotting | Santa Cruz |
| γH2ax | Mouse | 1:500, Immunofluorescence | Abcam |
| Sycp3 | Rabbit | 1:200, Immunofluorescence | Proteintech |

## Spectral domain-optical CT

Mice anesthetization and pupil dilation were conducted as above mentioned. Lateral images from nasal retina to temporal retina crossing through the optic nerve were collected using a SD-OCT system (OptoProbe, Burnaby, Canada). Thicknesses of retinal layers were measured by Photoshop software (Adobe, San Jose, CA, USA) in a double-blind manner.

## Immunofluorescence staining and analysis

Immunofluorescence staining was performed per a previously described protocol (*Jiang et al., 2016*; *Liu et al., 2015*). Mice eyecups and testicular tissues were enucleated after sacrifice, rinsed with 1×PBS with connective tissues trimmed, and fixed in 4% paraformaldehyde (PFA) at 4°C overnight. For eyecups, corneas and lens were subsequently removed without disturbing the retina. Posterior eyecups and testicular tissues were then dehydrated in 30% sucrose for 2 hr, embedded in optimal cutting temperature compound, and frozen sectioned at 5 μm. For in vitro assay, NIH3T3 cells were collected at 48 hr post transfection and fixed in 4% PFA at 4°C overnight. Retinal, testicular, and cellular sections were then blocked in 2% normal goat serum and permeabilized with 0.3% Triton X-100 at room temperature for 1 hr. Those sections were further incubated with primary antibodies (*Table 5*) at 4°C overnight and corresponding fluorescence-conjugated secondary antibodies (dilution: 1:1000; Invitrogen, Carlsbad, CA, USA) at room temperature for 1 hr. Nuclei were counterstained with Hoechst 33342 (Sigma, St. Louis, MO, USA). Retinal images were collected with a Leica TCS SP5 confocal system (Leica, Wetzlar, Germany), testicular images were taken by LSM 800 confocal

microscope (Carl Zeiss, Jena, Germany), and cellular cilia and centriole structures were photographed with Leica TCS SP8 super-resolution live cell imaging confocal system (Leica, Wetzlar, Germany). Leica LAS X Life Science Microscope Software (version 3.7.1.21655) was used to measure cilia and centriole length. ImageJ (version 2.0.0-rc-43/1.50e) was used to measure the relative mean fluorescence intensity of cone arrestin and Ift20 at centriole.

## 3D Structured illumination microscopy

To visualize Cep78, Ift20, and Ttc21a, NIH3T3 cells were stained with Cep78 (custom-made against antigen p457-741 of mouse Cep78 NP_932136.2), Ift20 (Proteintech), and Ttc21a (Abnova) antibodies (*Table 5*) for immunofluorescence. For SIM imaging, then photographed by an N-SIM S Super Resolution Microscope (Nikon, Tokyo, Japan) equipped with 405 nm, 488 nm, 561 nm, and 640 nm excitation light laser combiner (Coherent, Santa Clara, CA, USA), with CFI SR HP Apochromat TIRF 100XC Oil lens. The captured images were reconstructed with NIS-Elements AR (Advanced Research) Imaging Software (Nikon, Tokyo, Japan).

## TEM and SEM assays

Mice eyecups, mice semen cells collected from shredded unilateral caudal epididymis, mice testicular tissues, and human ejaculated sperm were subjected to TEM assay, respectively. Tissues were fixed in 2.5% glutaradehyde at 4°C overnight immediately after enucleation. Eyecups were dissected as described for TEM, and only the remaining posterior eyecups were used for the following experiments. Samples were then post-fixed in 1% osmic acid at 4°C for an hour, stained in aqueous 3% uranyl acetate for 2 hr, dehydrated with ascending concentrations of acetone, embedded in epoxy resin, cut into ultra-thin slides, and stained in 0.3% lead citrate. Ultrastructure of mice retina and human spermatozoa was visualized using a JEM-1010 electron microscope (JEOL, Tokyo, Japan). Ultrastructure of mice spermatids cells were observed with a Philips CM100 electron microscope (Philips, Amsterdam, North-Holland). Ultrastructure of testicular tissues was visualized using a FEI Tecnai G2 Spirit Bio TWIN electron microscope (Thermo, Waltham, MA, USA).

For SEM assay, mice sperms collected from shredded unilateral caudal epididymis and human ejaculated sperm were used. Human and mice spermatozoa were immersed in 2.5% glutaraldehyde at 4°C overnight, fixed in 1% osmic acid supplemented with 1.5% $K_3[Fe(CN)_3]$ at 4°C for an hour, steeped in 1% thiocarbohydrazide for an hour, post-fixed in 1% osmic acid at 4°C for an hour, and soaked in 2% uranyl acetate solution at 4°C overnight. The treated samples were then progressively dehydrated with an ethanol and isoamyl acetate gradient on ice, and dried with a $CO_2$ critical-point dryer (Eiko HCP-2, Hitachi Ltd., Tokyo, Japan). The specimens were subsequently mounted on aluminum stubs, sputter coated by an ionic sprayer meter (Eiko E-1020, Hitachi Ltd.), and visualized using a FEI Nova NanoSEM 450 SEM (Thermo, Waltham, MA, USA).

## CASA and sperm morphological study

Mice sperms, collected from shredded unilateral caudal epididymis, were maintained in 500 μL modified human tubal fluid (Irvine Scientific, Santa Ana, CA, USA) supplemented with 10% fetal bovine serum (Gibco, Grand Island, NY, USA) at 37°C for 7 min. Sperm concentration, motility, and progressive motility of semen samples were then assessed by IVOS II CASA system (Hamilton Thorne Inc, Beverly, MA, USA).

Human ejaculated semen sample was collected by masturbation after 7 days of sexual abstinence. Semen volume was measured with test tube. After liquefaction at 37°C for 30 min, semen sample was further subjected to BEION S3-3 CASA system (BEION, Beijing, China) for measurement of sperm concentration, motility, and progressive motility per the fifth edition of WHO guidelines.

Human and mice semen samples were stained on slides using the H&E staining method for sperm morphological analyses. At least 150 spermatozoa were analyzed for each group. Percentages of spermatozoa with normal morphology, abnormal head, abnormal neck, and abnormal flagella were calculated per the WHO guidelines. Abnormalities of sperm flagella were further classified into seven categories, including short flagella, absent flagella, coiled flagella, multi flagella, cytoplasm remains, irregular caliber, and angulation. Noteworthy, one spermatozoon was sorted into only one group based on its major flagellar abnormality.

## Histological H&E staining

For histological H&E staining, mice epididymal tissues and sperms collected from shredded unilateral caudal epididymis were rinsed with 1×PBS, fixed in 4% PFA at 4°C overnight, dehydrated with ascending concentrations of ethanol, embedded in paraffin, and sectioned into 5 µm slides. Slides were then deparaffinized in xylene, rehydrated with decreasing concentrations of ethanol, and stained with H&E for histological observation. Images were taken with a Zeiss Axio Skop plus2 microscope (ZEISS, Oberkochen, Germany).

## PAS staining

PAS staining was used for spermatogenic staging and evaluation of the development of seminiferous tubules. Fresh mice testicular tissues were fixed with modified Davidson's fluid fixation fluid at room temperature for 48 hr, embedded with paraffin, and sliced into sections. Slides were then dewaxed with xylene at 37°C for 30 min, rehydrated by descending concentrations of ethanol, dyed with PAS (Solarbio, Beijing, China) and hematoxylin in sequence, dehydrated with ascending concentrations of ethanol, and sealed with neutral balsam for observation. Images were taken with a Zeiss Axio Skop plus2 microscope (ZEISS, Oberkochen, Germany).

## Isolation of spermatid cells from testes

We isolated spermatid cells with STA-PUT method, following *Bryant et al., 2013*'s protocol. In brief, after removal of tunica vaginalis, testis tissue of *Cep78*[+/−] and *Cep78*[−/−] mice (n=6 for *Cep78*[+/−] and *Cep78*[−/−] *mice*) was digested with Collagenase IV (Gibco, RI, USA), Trypsin (Gibco, RI, USA), and DNaseI (Bomei, Hefei, China) in shaking water bath at 33°C for 5 min, filtered with 100 µm and 40 µm nylon filters (Corning, NY, USA), re-suspended in 0.5% bovine serum albumin (Sigma-Aldrich, St. Louis, MO, USA) solution, and settled in the BSA gradient in the sedimentation chamber of STA-PUT aparatus for 2 hr and 15 min. Cell fractions were obtained, stained by Hoechst 33342 (Sigma-Aldrich, St. Louis, MO, USA), and examined with a Zeiss Axio Observer A1 inverted microscope (ZEISS, Oberkochen, Germany) to assess the purity of spermatid cells.

## Custom-made antibody

The custom-made human CEP78 and mouse Cep78 antibodies were produced by Abclonal (Wuhan, China), p457-741 of mouse Cep78 (NP_932136.2) and p100-515 of human CEP78 protein (NP_001092272.1) were cloned to pET-28a (+) high-expression plasmid vectors, respectively. The immunogen was expressed in BL21 prokaryotic cells and immunized with experimental Japanese white rabbit. After the animals are sacrificed, the rabbit serum is subjected to antigen affinity purification.

## Sample preparation, IP, and MS analyses

Testicular tissues of *Cep78*[+/−] and *Cep78*[−/−] mice were lysed with Pierce IP lysis buffer (Thermo, Waltham, MA, USA) supplemented with 1% protease inhibitor cocktail 100× (Selleck Chemicals, Houston, TX, USA), respectively. Lysates were revolved for 1 hr at 4°C and centrifuged at 40,000 g for 1 hr. Supernatants were precleared with 20 µL of protein A/G magnetic beads (Millipore, Billerica, MA, USA) at 4°C for 1 hr. IP was performed using the Pierce co-IP kit (Thermo, Waltham, MA, USA). 5 µg of antibody or IgG was subjected for incubation per sample. For in vivo Cep78 IP, the custom-made mouse Cep78 antibody against antigen p457-741 of mouse Cep78 (NP_932136.2) was used. Eluted proteins were then subjected to SDS-PAGE, silver stain, immunoblotting, and MS. For MS, silver-stained gel lanes were carefully cut into small pieces and were then processed for MS analysis. Briefly, after trypsin digestion overnight, peptides were desalted using stage tips, re-suspended in 0.1% formic acid (v/v) and subjected to LC-MS (*Guo et al., 2011*; *Guo et al., 2010*; *Hu et al., 2013*; *Fan et al., 2020*).

For MS on elongating spermatids lysates of *Cep78*[+/−] and *Cep78*[−/−] mice, we isolated spermatid cells with STA-PUT. Lysed spermatid cells were digested overnight at 37°C with trypsin and subjected to the Tandem Mass Tag （TMT） labeling. For MS analyses, each fraction was analyzed using an Orbitrap Fusion Lumos mass spectrometer (Thermo Finnigan, San Jose, CA) coupled with Easy-nLC 1200 (Thermo Finnigan, San Jose, CA) and was separated by a High PH reverse phase separation microcapillary column (ACQUITY BEH C18 Column, 1.7 µm, 300 µm × 150 mm) at a flow rate of 4 µL/min in a 128 min linear gradient (3% buffer B for 14 min, 8% buffer B for 1 min, 29% buffer B

for 71 min, 41% buffer B for 12 min, 100% buffer B for 9 min, and 3% buffer B for 21 min; solvent A: 20 mM Ammonium formate, PH = 10; solvent B: 100% ACN, 0.1% FA). Sample were collected at a rate of one tube per minute, a total of 30 components. Easy1200 and fusion Lumos were used for series identification.

## co-IP assay

HEK293T cells from 100 mm dish were transfected (Vazyme, Nanjing, China) with 5 µg plasmid for each experimental group. Cells were collected at 48 hr post transfection and lysed with Pierce IP lysis buffer (Thermo, Waltham, MA, USA) supplemented with 1% protease inhibitor cocktail 100× (Selleck Chemicals, Houston, TX, USA). Lysates was revolved for 30 min at 4°C and centrifuged at 16,000 g for 30 min. Supernatants were incubated with 30 µL of Pierce anti-HA magnetic beads (Thermo, Waltham, MA, USA) or anti-DYKDDDDK IP resin (GenScript, Nanjing, China) at 4°C overnight. IP was performed using the Pierce co-IP kit (Thermo, Waltham, MA, USA), and eluted proteins were further subjected to immunoblotting.

## Size exclusion chromatography

2.43 mg of wild type C57BL/6J mouse testicular lysate extract was chromatographed with ÄKTA pure (GE Healthcare, Boston, MA, USA) modular chromatography system over a Superose 6 increase 10/300 GL column (Cytiva, Marlborough, MA, USA). The column was calibrated with gel filtration standard (Bio-Rad, Hercules, CA, USA) containing mixed markers of molecular weight ranging from 1.35 to 670 kDa. 1 mL of chromatography fractions was collected, and protein fractions were concentrated with refrigerated CentriVap vacuum concentrator (Labconco, Kansas City, MO, USA) and analyzed by western blot.

## RNA extraction and Q-PCR

NIH3T3 cells were harvested at 48 hr post transfection for RNA extraction. Total RNA was isolated from lysates of transfected cells using TRIzol reagent (Invitrogen, Carlsbad, CA, USA). RNA concentration and quality were determined with Nano-Drop ND-1000 spectrophotometer (Nano-Drop Technologies, Wilmington, DE, USA). cDNA was generated with a PrimeScript RT Kit (Takara, Otsu, Shiga, Japan). Q-PCR was conducted to detect RNA amounts using FastStart Universal SYBR Green Master (ROX; Roche, Basel, Switzerland) with StepOne Plus Real-Time PCR System (Applied Biosystems, Darmstadt, Germany). Primer sequences were listed in *Table 3*.

## Statistical analysis

GraphPad Prism (version 8.0; GraphPad Software, San Diego, CA, USA) was applied for statistical analyses. Student's t-test was utilized for comparisons between two different groups. One-way ANOVA followed by Dunnet's multiple comparison test was applied for comparisons among three or more groups. We presented data as mean ± SEM and considered $p < 0.05$ as statistically significant. All experiments were conducted in both biological and technical triplicates with data averaged.

## Acknowledgements

We thank all participants for their sample donations. This work was supported by the National Key R&D Program (2021YFC2700200 to XG); National Natural Science Foundation of China (82020108006, 81730025 to CZ, 81971439, 81771641 to XG, 82070974 to XC, 82060183 to XS, 82201764 to TZ); China Postdoctoral Science Foundation (2022M711676 to TZ); Shanghai Outstanding Academic Leaders (2017BR013 to CZ); and Six Talent Peaks Project in Jiangsu Province (YY-019 to XG); Basic Research Program of Jiangsu Province (BK20220316 to TZ); Scientific Research Project of Gusu School of Nanjing Medical University (GSBSHKY20213 to TZ). The funders had no role in study design, data collection and analysis, decision to publish, or preparation of the manuscript.

## Additional information

### Funding

| Funder | Grant reference number | Author |
|---|---|---|
| National Key Research and Development Program of China | 2021YFC2700200 | Xuejiang Guo |
| National Natural Science Foundation of China | 82020108006 | Chen Zhao |
| National Natural Science Foundation of China | 81730025 | Chen Zhao |
| National Natural Science Foundation of China | 81971439 | Xuejiang Guo |
| National Natural Science Foundation of China | 81771641 | Xuejiang Guo |
| National Natural Science Foundation of China | 82070974 | Xue Chen |
| National Natural Science Foundation of China | 82060183 | Xunlun Sheng |
| National Natural Science Foundation of China | 82201764 | Tianyu Zhu |
| China Postdoctoral Science Foundation | 2022M711676 | Tianyu Zhu |
| Shanghai Outstanding Academic Leaders | 2017BR013 | Chen Zhao |
| Six Talent Peaks Project in Jiangsu Province | YY-019 | Xuejiang Guo |
| Basic Research Program of Jiangsu Province | BK20220316 | Tianyu Zhu |
| Scientific Research Project of Gusu School of Nanjing Medical University | GSBSHKY20213 | Tianyu Zhu |

The funders had no role in study design, data collection and interpretation, or the decision to submit the work for publication.

### Author contributions

Tianyu Zhu, Conceptualization, Resources, Data curation, Software, Formal analysis, Funding acquisition, Validation, Investigation, Visualization, Writing – original draft, Writing – review and editing, Designed experiments in Figure 3-7; Yuxin Zhang, Conceptualization, Resources, Data curation, Software, Formal analysis, Validation, Investigation, Visualization, Writing – original draft, Writing – review and editing, Wrote the paper; Xunlun Sheng, Conceptualization, Resources, Data curation, Supervision, Funding acquisition, Validation, Investigation, Visualization, Project administration, Writing – review and editing, Clinical information collection and sample management; Xiangzheng Zhang, Resources, Data curation, Software, Formal analysis, Investigation, Writing – review and editing, Performed mass spectrometry experiments and analysis; Yu Chen, Data curation, Formal analysis, Investigation, Writing – review and editing, Performed CASA analysis; Hongjing Zhu, Data curation, Software, Formal analysis, Validation, Investigation, Visualization, Writing – review and editing, Performed Nphp1 and cone-arresting immunofluorescence; Yueshuai Guo, Resources, Data curation, Software, Formal analysis, Supervision, Validation, Investigation, Visualization, Writing – review and editing, supervised mass spectrometry experiments and analysis; Yaling Qi, Supervision, Validation, Investigation, Methodology, Writing – review and editing, Supervised lentivirus transfection experiment; Yichen Zhao, Formal analysis, Validation, Investigation, Writing – review and editing, Performed Ift20 in vivo immunoprecipitation and western blot of the immunoprecipitation and gel filtration samples; Qi Zhou, Data curation, Software, Formal analysis, Validation, Investigation, Visualization,

Writing – review and editing, Performed mice genotyping; Xue Chen, Conceptualization, Resources, Data curation, Software, Formal analysis, Supervision, Funding acquisition, Validation, Investigation, Visualization, Methodology, Writing – original draft, Project administration, Writing – review and editing, Wrote the paper; Xuejiang Guo, Conceptualization, Resources, Data curation, Software, Formal analysis, Supervision, Funding acquisition, Validation, Investigation, Visualization, Methodology, Writing – original draft, Project administration, Writing – review and editing, Wrote the paper; Chen Zhao, Conceptualization, Resources, Data curation, Software, Formal analysis, Supervision, Funding acquisition, Validation, Investigation, Visualization, Methodology, Writing – original draft, Project administration, Writing – review and editing, Wrote the paper

### Author ORCIDs
Yaling Qi http://orcid.org/0000-0002-2940-4733
Xuejiang Guo http://orcid.org/0000-0002-0475-5705
Chen Zhao http://orcid.org/0000-0003-1373-7637

### Ethics

Human subjects: Our study, conformed to the Declaration of Helsinki, was prospectively reviewed, and approved by the ethics committee of People's Hospital of Ningxia Hui Autonomous Region ([2016] Ethic Review [Scientific Research] NO. 018) and Nanjing Medical University (NMU Ethic Review NO. (2019) 916). Signed informed consents were obtained from all individuals in the study.

All mice were raised in a specific-pathogen-free animal facility accredited by Association for Assessment and Accreditation of Laboratory Animal Care (AAALAC) in Model Animal Research Center, Nanjing University, China. The facility provided ultraviolet sterilization, a 12-hour light/dark cycle, ad libitum access to water, and standard mouse chow diet. Mice experiments were performed in accordance with approval of the Institutional Animal Care and Use Committee of Nanjing Medical University (IACUC-1707017-8) and with the ARVO Statement for the Use of Animals in Ophthalmic and Vision Research.

### Decision letter and Author response
Decision letter https://doi.org/10.7554/eLife.76157.sa1
Author response https://doi.org/10.7554/eLife.76157.sa2

---

## Additional files

### Supplementary files
• Supplementary file 1. Protein groups and cellular components analysis of mass spectrometry (MS) proteomics on elongating spermatids lysates of $Cep78^{+/-}$ and $Cep78^{-/-}$ mice. (A) Different expression proteins between $Cep78^{+/-}$ and $Cep78^{-/-}$ male mice elongating spermatids. (B) The enriched cellular components of different expression proteins between $Cep78^{+/-}$ and $Cep78^{-/-}$ male mice elongating spermatids.

• Transparent reporting form

### Data availability

All H&E, PAS, immunofluorescence, TEM, SEM, uncropped gels and blots, and statistical data are available at corresponding source data files. All mass spectrometry data are available at Dryad.

The following datasets were generated:

| Author(s) | Year | Dataset title | Dataset URL | Database and Identifier |
|---|---|---|---|---|
| Zhu T, Zhang Y, Sheng X, Zhang X, Chen Y, Guo Y, Qi Y, Chen X, Guo X, Zhao C | 2023 | Anti-Cep78 immunoprecipitation (IP) coupled with quantitative MS (IP-MS) on testicular lysates of Cep78+/- and Cep78-/- mice | http://doi.org/10.5061/dryad.6djh9w12z | Dryad Digital Repository, 10.5061/dryad.6djh9w12z |

*Continued on next page*

*Continued*

| Author(s) | Year | Dataset title | Dataset URL | Database and Identifier |
|---|---|---|---|---|
| Zhu T, Zhang Y, Sheng X, Zhang X, Chen Y, Guo Y, Qi Y, Chen X, Guo X, Zhao C | 2023 | Quantitative mass spectrometry (MS) on elongating spermatids lysates of Cep78+/- and Cep78-/- mice | http://doi.org/10.5061/dryad.stqjq2c4p | Dryad Digital Repository, 10.5061/dryad.stqjq2c4p |

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
