## [Editor Report]

This paper is of interest to scientists within the cilia and centrosome fields, especially those studying photoreceptor and sperm development and the diseases associated with their dysfunction. The phenotypic characterization Cep78-/- mutant mice, revealing severe structural and functional defects of photoreceptors and sperm flagella, is convincing and consistent with recently published work. While CEP78 is shown to physically interact with IFT20 and TTC21A, the precise molecular mechanisms by which CEP78 affects photoreceptor and sperm development remain to be clarified.

---

## [Decision Letter]

**Decision letter after peer review:**

Thank you for submitting your article "Study on gene knockout mice and human mutant individual reveals absence of CEP78 causes photoreceptor and sperm flagella impairments" for consideration by *eLife*. Your article has been reviewed by 3 peer reviewers, one of whom is a member of our Board of Reviewing Editors, and the evaluation has been overseen by Ricardo Azziz as the Senior Editor. The reviewers have opted to remain anonymous.

Essential revisions:

1) The claim that CEP78 binds directly to IFT20 and TTC21A is not justified by the data provided. Either this data should be removed from the manuscript or additional experiments (including relevant negative control IPs) should be performed to support this claim. For example, the IPs of mouse testis extracts that were used for the MS analysis in Table S4 were performed with an antibody against endogenous CEP78. One caveat with this approach is that the antibody might block binding of CEP78 to some of its interactors, e.g. if the epitope recognized by the antibody is located within one or more interactor binding sites in CEP78. This could explain why the authors did not identify some of the previously identified CEP78 interactors in their IP, such as CEP76 and the EDD-DYRK2-DDB1-VprBP complex (Hossain et al. 2017) as well as CEP350 (Goncalves et al. 2021).

2) Details about what part of the CEP78 protein the custom-made antibody detects (before or after 10 bp deletion?), as well as how it was made, need to be provided. This would be necessary to know in order to eliminate the possibility of the existence of truncated CEP78 protein in the affected patient, and could also help clarify whether the antibody might inhibit binding of CEP78 to previously identified binding partners (see comment above).

3) Figure 1F and Figure 4K: the data needs to be quantified.

4) In Figure 2A, another marker (e.g. NPHP1) for the connecting cilium should be used, as the connecting cilium is not apparent in the images shown here. Also, notation for red arrows in Figure 2 should be provided, and authors should comment on the apparent shape change of the connecting cilium in the mutant, which seems disorganized and swollen compared to control.

5) The efficiency of the siRNA knockdown shown in 7J-M was only assessed by qPCR (Figure S4), but this does not necessarily mean the corresponding proteins were depleted. Western blot analysis needs to be performed to show depletion at the protein level. Furthermore, it would be desirable with rescue experiments to validate the specificity of the siRNAs used.

6) Figure 7I: the resolution of the IFM is not very high and certainly not sufficient to demonstrate that CEP78, IFT20 and TTC21A co-localize to the same region on the centrosome, which one would have expected if they directly interact. CEP78 was previously shown to localize to the distal end of the (mother) centriole wall in cultured mammalian cells (Brunk et al. 2016; Hossein et al., 2017; Goncalves et al., 2021), whereas IFT proteins are concentrated near the transition fibers. If authors want to claim direct interaction and co-localisation between these proteins, higher resolution images (e.g. IEM, STED, 3D-SIM or similar) of these proteins in spermatids need to be provided. Such an analysis might also provide useful information regarding the mechanism by which CEP78 affects flagellum biogenesis, and possibly explain why the mechanism appears different from that observed in cultured human cells (Figure S5, Discussion lines 467-471).

7) The manuscript text needs to be rewritten to improve clarity, eliminate grammatical errors, make several statements more accurate, and better present and discuss the results in the context of known literature in the field. Specifically:

a) It is not immediately clear if the human mutation was identified by the authors and presented for the first time in this publication or was already done before. 'In this study, based on results of a male patient carrying CEP78 mutation and Cep78 gene knockout mice, we report CEP78 as a new causative gene for a distinct syndrome involving two phenotypes, CRD and male sterility[16].' Reference 16 does not seem related to the discovery of the CEP78 mutation and it is unclear why it is included here (lanes 106-108). If the patient's CEP78 mutation is the same as published in Qing Fu et al. (10.1136/jmedgenet-2016-104166) why it was reported as causing Usher syndrome and in the current publication it is not mentioned at all? Do authors think that CEP78 mutation is associated with Usher's syndrome? It is important to discuss this.

b) In a previous study a missense variant in CEP78 was shown to be associated with astenoteratozoospermia and reduced male fertility, but no morphological defects in sperm were observed (Ascari, G., et al., 2020). The authors should discuss this obvious phenotypic and functional difference between the missense variant and 10 bp deletion in CEP78, that they have studied. Also, they may want to comment on the low penetrance of the infertility phenotype in human patients (e.g. see discussion on this in PMID: 35240912).

c) Page 5, given the previous report of CEP78 patients with retina degeneration, hearing loss, and reduced infertility (Ascari et al. 2020), the statement of "we report CEP78 as a NEW causative gene for a distinct syndrome…TWO phenotypes….." Is not accurate.

d) Introduction. The statement that "CRD usually exists with combination of immotile cilia defects in other systems" is not correct. CRD due to ciliopathy can have cilia-related syndromic defects in other systems but it is a relatively small portion of all CRDs and the most frequently mutated genes are not cilia-related genes, such as ABCA4, GUCY2D, CRX.

*Reviewer #1 (Recommendations for the authors):*

1) The manuscript contains several grammatical errors and would benefit from proofreading by an expert in English language.

2) Reference list: references 8 and 23 are identical; reference 19 is incomplete.

3) Figures: the following figures are very small, and it is difficult to see all the details: Figure 1F, H, I, J; Figure 4C-J; Figure5N. I recommend increasing the figure size.

4) Line 27: fertility should be infertility.

5) Line 92: I believe CEP78 has more than 2 leucine rich repeats (up to 7 or 8), which are distributed in 2 leucine-rich repeat regions. Please double-check and clarify.

6) Line 180: CEP78 is not a ciliary protein, it is a centrosomal protein. Please correct.

7) Table S4 and Methods section page 34: the authors need to indicate which CEP78 antibody was used for the IP analysis and how much of it was used.

8) CEP78 was previously shown to localize to the distal end of the (mother) centriole wall in cultured mammalian cells (Brunk et al. 2016; Hossein et al., 2017; Goncalves et al., 2021), whereas IFT proteins are concentrated near the transition fibers. To support the authors' claim that CEP78 forms a trimeric complex with IFT20 and TTC21A they should consider doing high-resolution localization analysis (TEM, STED, 3D-SIM or similar) of these proteins in spermatids. Such an analysis might also provide useful information regarding the mechanism by which CEP78 affects flagellum biogenesis, and possibly explain why the mechanism appears different from that observed in cultured human cells (Figure S5, Discussion lines 467-471).

9) Figure S5, Table S2 and Table S3: CP110 looks downregulated in the testis of the Cep78-/- animals, yet CP110 was not detected in the MS analysis shown in Tables S2 and S3. Please explain this discrepancy.

10) Methods section: information describing how the CEP78 antibodies were produced is missing.

*Reviewer #2 (Recommendations for the authors):*

1. I suggest rewriting the introduction section to make the statement accurate.

2. Better quantify the data as suggested in my specific comments.

3. Conduct experiments suggested to provide evidence to support the conclusion. To show the three protein form timer, additional experimental evidence such as gel filtration is needed. To better visualize CC, NPHP1 or other proper markers should be used.

*Reviewer #3 (Recommendations for the authors):*

The experimental base is sufficient in this research to validate the statements. However, writing improvement and coherence in presentation would help to improve the readability significantly.

Following are suggestions for manuscript improvement:

– It was not immediately clear if human mutation was identified by the authors and presented for the first time in this publication or was already done before. 'In this study, based on results of a male patient carrying CEP78 mutation and Cep78 gene knockout mice, we report CEP78 as a new causative gene for a distinct syndrome involving two phenotypes, CRD and male sterility[16].' I have followed reference 16 to understand that it is not related to discovery of CEP78 mutation. It is unclear why reference 16 is included (lanes 106-108).

– Missing clear information if the patient/-s are the same as published in Qing Fu et al., (10.1136/jmedgenet-2016-104166).

– If the patient's CEP78 mutation is the same as published in Qing Fu et al., (10.1136/jmedgenet-2016-104166) why it was reported as causing Usher syndrome and in current the publication it is not mentioned at all? Do authors think that CEP78 mutation is associated with Usher's syndrome? It is important to discuss this.

– The missense variant in CEP78 was shown to be associated with astenoteratozoospermia and reduced male fertility but, no morphological defects were found in sperm (Ascari, G., et al., 2020). Authors should consider discussing this obvious phenotypic and functional difference between missense variant and 10 bp deletion in CEP78, that they have studied. Link to IP-MS data submitted to Dryad, Dataset, (https://doi.org/10.5061/dryad.6djh9w12z) was not working during the revision process (Error message: link incorrect or not activated yet).

– What part of the CEP78 protein antibody detects (before or after 10 bp deletion)? This would be interesting to know in order to eliminate the possibility of the existence of the truncated CER78 protein in the affected patients.

– Authors refer to the same publication but use different reference numbers in the text and in the reference list: reference: nr 8 Fu, Q., et al., CEP78 is mutated in a distinct type of Usher syndrome. J Med Genet, 2017. 54(3): p. 190-195. nr 23: Fu, Q., et al., CEP78 is mutated in a distinct type of Usher syndrome. J Med Genet, 2017. 54(3): p. 190-195. Please address this.

– Gene should be italic: lines 153, 177, 227, 233, 240, 246, 273, 274,278, 279 and oth.

– Line 271; deletion of gene not protein.

– English language revision would improve readability.

---

## [Author Response]

Essential revisions:1) The claim that CEP78 binds directly to IFT20 and TTC21A is not justified by the data provided. Either this data should be removed from the manuscript or additional experiments (including relevant negative control IPs) should be performed to support this claim. For example, the IPs of mouse testis extracts that were used for the MS analysis in Table S4 were performed with an antibody against endogenous CEP78. One caveat with this approach is that the antibody might block binding of CEP78 to some of its interactors, e.g. if the epitope recognized by the antibody is located within one or more interactor binding sites in CEP78. This could explain why the authors did not identify some of the previously identified CEP78 interactors in their IP, such as CEP76 and the EDD-DYRK2-DDB1-VprBP complex (Hossain et al. 2017) as well as CEP350 (Goncalves et al. 2021).

Thanks for this insightful suggestion. Firstly, the antigenic sequence of our Cep78 antibody is p457-741 (NP_932136.2). Cep78 was reported to bind DD-DYRK2-DDB1-VprBP complex, the 1-520aa is responsible for Cep78’s interaction with VprBP, and deletion of p450-497 didn’t affect Cep78’s interaction with VprBP, indicating1-450aa of Cep78 might interact with VprBp (Hossain et al. 2017). Our anti-Cep78 antibody recognizes p457-741 of Cep78, the binding of Cep78 (p1-457aa) to VprBP is not expected to be blocked by our anti-Cep78 antibody. However, VprBp was not detected by our IP-MS experiment. C-terminal region (395-722aa) of Cep78 overlaps with our Cep78 antibody’s antigenic region (p457-741), and interacts with Cep350 (Goncalves et al. 2021). As a polyclonal antibody, our anti-Cep78 antibody seems not to block the interaction of Cep78 (p457-741) with other proteins, because we still identified Cep350 in our IP-MS. So we believe that our Cep78 antibody may not block the potential interaction of endogenous Cep78 with other known interactors.

Ttc21a and Ift20 were identified as Cep78’s interacting protein by pulling down endogenous Cep78. To further validate the interaction between Cep78 and Ttc21a or Ift20, we performed reciprocal co-IP between Cep78 and Ttc21a or Ift20 by overexpression (Figure 7A-C), and observed colocalization between Cep78 and Ttc21a or Ift20. As suggested by the reviewer, we added Gapdh (Figure 7D) and Ap80-NB-HA (Figure 7—figure supplement 2 A-C) in co-IP as negative controls. Besides, we provided evidence that Cep78, Ift20 and Ttc21a form a complex, as they all co-fractioned in a testicular protein complex at the size between158 kDa to 670 kDa using size exclusion chromatography (Figure 7E). With these data, we think that Cep78 interacts with Ttc21a and Ift20. We rephrased “direct interaction” as “interaction” in the manuscript.

We believe that our findings of interactions of Cep78 with Ttc21a or Ift20 do not contradict the previously reports of the functions of Cep78. And we believe that our findings provide novel molecular basis of Cep78 in sperm flagella formation.

2) Details about what part of the CEP78 protein the custom-made antibody detects (before or after 10 bp deletion?), as well as how it was made, need to be provided. This would be necessary to know in order to eliminate the possibility of the existence of truncated CEP78 protein in the affected patient, and could also help clarify whether the antibody might inhibit binding of CEP78 to previously identified binding partners (see comment above).

Thanks for reviewer’s comments. CEP78 antibody was generated based on the antigen sequence of p100-515 in CEP78 protein (NP_001092272.1). And the antigen sequence is before the 10 bp deletion in the patient with mutation of canonical splicing acceptor site of exon 14 (c.1629-2A>G, p.?). If truncated CEP78 (p.G545Pfs*6) protein of 61 kD is expressed, it will contain the antigen sequence and is expected to be detected by our anti-CEP78 antibody. Our Western blot results showed no expression of truncated CEP78 in the patient (Figure 6—figure supplement 1).

We have added the following detail information of the antibody in the section of Materials and methods: The custom-made human CEP78 and mouse Cep78 antibodies were produced by Abclonal (Wuhan, China), p457-741 of mouse Cep78 (NP_932136.2) and p100-515 of human CEP78 protein (NP_001092272.1) were cloned to pET-28a (+) high-expression plasmid vectors, respectively. The immunogen was expressed in BL21 prokaryotic cells and immunized with experimental Japanese white rabbit. After the animals are sacrificed, the rabbit serum is subjected to antigen affinity purification. Which is at Page 15-16, line 341-348, line 357-358 in our new version of manuscript.

3) Figure 1F and Figure 4K: the data needs to be quantified.

Thank you for this suggestion. For Figure 4K, we stained *Cep78^+/-^* and *Cep78^-/-^* spermatids with anti-Centrin 1 to measure the centriole length. The statistical data of centriole length were provided (Figure 4L), showing significantly increased centriole lengths in *Cep78^-/-^*spermatids.

For Figure 1F, we have quantified the immunofluorescence intensity of cone arrestin in light-adapted retinas of *Cep78^+/-^* and *Cep78^-/-^* mice at 3-month. The results indicate that immunofluorescence intensity of the cone arrestin was lower in *Cep78^-/-^* mice.

4) In Figure 2A, another marker (e.g. NPHP1) for the connecting cilium should be used, as the connecting cilium is not apparent in the images shown here. Also, notation for red arrows in Figure 2 should be provided, and authors should comment on the apparent shape change of the connecting cilium in the mutant, which seems disorganized and swollen compared to control.

Thank you for your suggestion, we have stained retinal cryosections from *Cep78^+/-^* and *Cep78^-/-^* mice with anti-Nphp1 to visualize connecting cilium (Figure 2A-B). The results indicated that connecting cilia are shortened in *Cep78^-/-^* mice compared to *Cep78^+/-^* (Figure 2A-B). We also re-measured the length of CC as the length of ciliary structure out of outer segment in TEM (Figure 2C-D). The ciliary structure inside the outlet segment is axoneme, which is not included in the length measurement. And our TEM results showed shortened connecting cilia in photoreceptors of *Cep78^-/-^* mice (Figures 2C-D), which is consistent with the results of Nphp1immunofluorescence.

Besides, we observed that upper parts of connecting cilia were swelled with disorganized microtubules in TEM (Figure 2E-G). The red arrows in Figure 2E-G indicated swelled upper part of connecting cilia and disorganized microtubules of *Cep78^-/-^* photphoreceptors.

5) The efficiency of the siRNA knockdown shown in 7J-M was only assessed by qPCR (Figure S4), but this does not necessarily mean the corresponding proteins were depleted. Western blot analysis needs to be performed to show depletion at the protein level. Furthermore, it would be desirable with rescue experiments to validate the specificity of the siRNAs used.

Thank you for this suggestion. To validate the specificity of the siRNAs used, we performed rescue experiments using rescue plasmid with siRNA targeting sequence synonymously mutated (Table 4).

The efficiency of siRNA knockdown were evaluated by both qPCR (Figure 7—figure supplement 3.A-C) and Western Blot (Figures 7.J-K, Figure 7—figure supplement 3.D-E,H-I). The results showed that siRNAs significantly reduced the expression of Cep78, Ift20, and Ttc21a at both mRNA Figure 7—figure supplement 3.A-C and protein levels (Figures 7.J-K, Figure 7—figure supplement 3.A-C), respectively. Meanwhile, co-transfection of rescue plasmids of Cep78^syn^-HA, Ift20^syn^-Flag and Ttc21a^syn^-Flag rescued the expression of Cep78, Ift20, and Ttc21a at both mRNA (Figure 7—figure supplement 3.A-C) and protein levels (Figures 7.J-K, Figure 7—figure supplement 3.D-E, H-I) compared with siRNA groups, respectively.

Additionally, we evaluated whether the effects are specific for Cep78, Ift20 and Ttc21siRNAs in the regulation of cilia and centriole lengths. The results showed that the suppression effects of Cep78, Ift20 and Ttc21 siRNAs on cilia and centriole lengths could be rescued by the overexpression of rescue plasmids of Cep78^syn^-HA, Ift20^syn^-Flag and Ttc21a^syn^-Flag (Figures 7.N-S), respectively.

6) Figure 7I: the resolution of the IFM is not very high and certainly not sufficient to demonstrate that CEP78, IFT20 and TTC21A co-localize to the same region on the centrosome, which one would have expected if they directly interact. CEP78 was previously shown to localize to the distal end of the (mother) centriole wall in cultured mammalian cells (Brunk et al. 2016; Hossein et al., 2017; Goncalves et al., 2021), whereas IFT proteins are concentrated near the transition fibers. If authors want to claim direct interaction and co-localisation between these proteins, higher resolution images (e.g. IEM, STED, 3D-SIM or similar) of these proteins in spermatids need to be provided. Such an analysis might also provide useful information regarding the mechanism by which CEP78 affects flagellum biogenesis, and possibly explain why the mechanism appears different from that observed in cultured human cells (Figure S5, Discussion lines 467-471).

Thank the reviewer for the constructive suggestions. To better demonstrate co-localization of CEP78, IFT20 and TTC21A on the centrosome, we overexpressed Cep78-Halo, Ift20-mCherry and Ttc21a-mEmerald in NIH3T3 cells by lentivirus, and photographed super-resolution images with SIM (N-sim, Nikon, Tokyo, Japan). The SIM results showed that Ift20 and Ttc21a co-localized with Cep78 (Figure 7F). Cep78 was previously reported to localize at the centriole (Goncalves et al., 2021). The co-localization of Cep78, Ift20 and Ttc21a indicated possible important roles of Cep78 in the regulation of Ift20 and Ttc21a in centriole. Our interaction analysis revealed that Cep78 interacted with Ift20 and Ttc21a (Figure 7D), and formed a complex with Ift20 and Ttc21a (Figure 7E). Loss of Cep78 down-regulated the expression of and interaction between Ift20 and Ttc21a (Figures 7G-M). Loss of either of Cep78, Ift20 and Ttc21a caused ciliogenesis defects (Figures 7N-O), and loss of either Cep78 or Ift20 caused abnormal centriole elongation (Figures 7P-Q). Previous studies have shown that depletion of *Ift20* (PMID: 27682589) or *Ttc21a* (PMID: 30929735) caused sperm flagella and sperm head shaping defects, similar to the phenotypes of *Cep78* knockout sperm.

7) The manuscript text needs to be rewritten to improve clarity, eliminate grammatical errors, make several statements more accurate, and better present and discuss the results in the context of known literature in the field. Specifically:a) It is not immediately clear if the human mutation was identified by the authors and presented for the first time in this publication or was already done before. 'In this study, based on results of a male patient carrying CEP78 mutation and Cep78 gene knockout mice, we report CEP78 as a new causative gene for a distinct syndrome involving two phenotypes, CRD and male sterility[16].' Reference 16 does not seem related to the discovery of the CEP78 mutation and it is unclear why it is included here (lanes 106-108). If the patient's CEP78 mutation is the same as published in Qing Fu et al. (10.1136/jmedgenet-2016-104166) why it was reported as causing Usher syndrome and in the current publication it is not mentioned at all? Do authors think that CEP78 mutation is associated with Usher's syndrome? It is important to discuss this.

Thank the reviewer for pointing this out. Reference 16 was not correctly cited, during revision we have removed this citation. We apologize for such mistake.

The patient studied in this manuscript is the same as that reported in publication by Qing Fu et al. (10.1136/jmedgenet-2016-104166), thus, we added “We have previously linked the *CEP78* c.1629-2A>G mutation to autosomal recessive CRDHL, and revealed a 10 bp deletion of *CEP78* exon 14 in mRNA of white blood cells from the patient carrying *CEP78* c.1629-2A>G mutation8[]. We further explored whether c.1629-2A>G mutation in this previously visited patient would disturb CEP78 protein expression and male fertility.”.

In Qing Fu et al. ’s study, we termed CEP78 mutation caused disease “a distinct type of Usher syndrome”, because patients with CEP78 mutation have syndromes similar but not identical to Usher syndrome. CEP78 disruption distinctively affects cone photoreceptors earlier, while in all other Usher syndrome types, rod photoreceptor functions are initially compromised (PMID: 11701652). This indicates CEP78 mutation is associated with a distinct type of Usher syndrome. So, with revisit of this patient, we found it is more accurate to describe the patient’s symptoms with “cone rod dystrophy”. So, in our new version of manuscript, we report this patient is associate with cone rod dystrophy. We hope our explanation can reduce the misunderstandings.

b) In a previous study a missense variant in CEP78 was shown to be associated with astenoteratozoospermia and reduced male fertility, but no morphological defects in sperm were observed (Ascari, G., et al., 2020). The authors should discuss this obvious phenotypic and functional difference between the missense variant and 10 bp deletion in CEP78, that they have studied. Also, they may want to comment on the low penetrance of the infertility phenotype in human patients (e.g. see discussion on this in PMID: 35240912).

Thank the reviewer for the suggestion. In this study, we reported a patient with *CEP78* c.1629-2A>G mutation, which caused homozygous 10 bp deletion of CEP78 exon 14 in mRNA, and CEP78 protein absence. Mutation of CEP78 caused complete absence of CEP78 protein. We found this patient has male infertility and multiple morphological abnormalities of the sperm flagella (MMAF) with oligoasthenoteratozoospermia.

Ascari, G., et al. reported 3 patients with male reproductive phenotypes, among whom only one patient had mutations only in *CEP78* gene and absence of its full length protein expression. The patient had compound heterozygous mutations with a missense and a nonsense mutation (c.[449T>C];[1462-1G>T] p.[Leu150Ser];[?]), and exhibited asthenoteratozoospermia and male infertility, with the sperm phenotype less severe than that observed in our study. The patient in our study showed phenotype of oligoasthenoteratozoospermia with MMAF, consistent with the phenotype observed in Cep78 knockout mice. Whether the patient reported by Ascari, G., et al. had truncated CEP78 by nonsense mutation was not analyzed, which might contribute the phenotypic differences between patients.

The above is discussed at Page 35,Line 769-779 in the revised manuscript.

c) Page 5, given the previous report of CEP78 patients with retina degeneration, hearing loss, and reduced infertility (Ascari et al. 2020), the statement of "we report CEP78 as a NEW causative gene for a distinct syndrome…TWO phenotypes….." Is not accurate.

Thank the reviewer for the comments. We have removed the term “NEW causative gene” in Page 5, Line 108 of the revised version of our manuscript. The revised sentence is “In this study, based on results of a male patient carrying *CEP78* mutation and *Cep78* gene knockout mice, we report *CEP78* as a causative gene for CRD and male sterility.”

d) Introduction. The statement that "CRD usually exists with combination of immotile cilia defects in other systems" is not correct. CRD due to ciliopathy can have cilia-related syndromic defects in other systems but it is a relatively small portion of all CRDs and the most frequently mutated genes are not cilia-related genes, such as ABCA4, GUCY2D, CRX.

Thank the reviewer for the comments. We agree with the reviewer that only a small portion of CRDs are due to cilia defects and can have cilia-related syndromic defects in other systems. We corrected this statement in Page 4, Line 80-82 of the revised version of our manuscript. In our revised version, the statement has been changed to “A small portion of CRDs are due to retina cilia defects, and they may have cilia-related syndromic defects in other systems[1].”

Reviewer #1 (Recommendations for the authors):1) The manuscript contains several grammatical errors and would benefit from proofreading by an expert in English language.

Thank you for this suggestion. We have carefully checked the grammar and spelling, we hope that the new version of the article has a better reading experience.

2) Reference list: references 8 and 23 are identical; reference 19 is incomplete.

Thank the reviewer for the reminding. We have removed the identical reference 23, and completed information of reference 18 (original reference 19) at Page 42, Line 918-919 in the revised manuscript.

3) Figures: the following figures are very small, and it is difficult to see all the details: Figure 1F, H, I, J; Figure 4C-J; Figure5N. I recommend increasing the figure size.

Thank the reviewer for the suggestions. We have enlarged the above-mentioned figures in Figure 1, 4 and 5 to better show the details,.

4) Line 27: fertility should be infertility.

Thank the reviewer for the comment. We have corrected “fertility” to “infertility” in page 2, Line 27 in the revised manuscript.

5) Line 92: I believe CEP78 has more than 2 leucine rich repeats (up to 7 or 8), which are distributed in 2 leucine-rich repeat regions. Please double-check and clarify.

Thank the reviewer for the suggestion. We have checked the NCBI gene database, and found that CEP78 protein had six consecutive leucine-rich repeats (LRR) in 2 leucine-rich repeat regions (https://www.ncbi.nlm.nih.gov/gene/84131). To avoid confusion, we modified “Centrosomal protein of 78 kDa (CEP78), protein encoded by the *CEP78* gene (MIM: 617110), is a centriolar protein composed of 722 amino acids and possesses two leucine-rich repeats and a coiled-coil domain[5].” to “Centrosomal protein of 78 kDa (CEP78), protein encoded by the *CEP78* gene (MIM: 617110), is a centriolar protein composed of 722 amino acids and possesses two leucine-rich repeat regions and a coiled-coil domain[5]” in Page 4, Line 92-94 in the revised manuscript.

6) Line 180: CEP78 is not a ciliary protein, it is a centrosomal protein. Please correct.

Thank the reviewer for the comment. According to the suggestion, we have corrected “Since CEP78 is a ciliary protein and *CEP78* mutation was found associated with primary-cilia defects.” to “Since CEP78 is a centrosome protein important for ciliogenesis and *CEP78* mutation was found associated with primary-cilia defects” at Page 22, Line 485-486 in the revised manuscript.

7) Table S4 and Methods section page 34: the authors need to indicate which CEP78 antibody was used for the IP analysis and how much of it was used.

Thank the reviewer for the suggestions. To clearly addressed the amount and the type of CEP78 antibody for IP analysis we added “5ug of antibody or IgG was subjected for incubation per sample. For in vivo Cep78 IP, the custom-made mouse Cep78 antibody against antigen p457-741 of mouse Cep78 (NP_932136.2) was used. ” in Materials and methods section, which is in Page 16, Line 357-360in the revised manuscript. We also added “The antibody against mouse Cep78 (p457-741) was used for IP-MS analysis.” in Table 2 in the revised manuscript.

8) CEP78 was previously shown to localize to the distal end of the (mother) centriole wall in cultured mammalian cells (Brunk et al. 2016; Hossein et al., 2017; Goncalves et al., 2021), whereas IFT proteins are concentrated near the transition fibers. To support the authors' claim that CEP78 forms a trimeric complex with IFT20 and TTC21A they should consider doing high-resolution localization analysis (TEM, STED, 3D-SIM or similar) of these proteins in spermatids. Such an analysis might also provide useful information regarding the mechanism by which CEP78 affects flagellum biogenesis, and possibly explain why the mechanism appears different from that observed in cultured human cells (Figure S5, Discussion lines 467-471).

Thank the reviewer for the constructive suggestions. To better demonstrate co-localization of CEP78, IFT20 and TTC21A on the centrosome, we overexpressed Cep78-Halo, Ift20-mCherry and Ttc21a-mEmerald in NIH3T3 cells by lentivirus, and photographed super-resolution images with SIM (N-sim, Nikon, Tokyo, Japan). The SIM results showed that Ift20 and Ttc21a co-localized with Cep78 (Figure 7F). Cep78 was previously reported to localize at the centriole (Goncalves et al., 2021). The co-localization of Cep78, Ift20 and Ttc21a indicated possible important roles of Cep78 in the regulation of Ift20 and Ttc21a in centriole. Our interaction analysis revealed that Cep78 interacted with Ift20 and Ttc21a (Figure 7D), and formed a complex with Ift20 and Ttc21a according to the fractionation analysis by size exclusion chromatography (Figure 7E). Loss of Cep78 down-regulated the expression of and interaction between Ift20 and Ttc21a (Figures 7G-M).

9) Figure S5, Table S2 and Table S3: CP110 looks downregulated in the testis of the Cep78-/- animals, yet CP110 was not detected in the MS analysis shown in Tables S2 and S3. Please explain this discrepancy.

Thank the reviewer for the comment. In our MS analysis, we identified 7505 proteins in mouse spermatids. Although the number of identified proteins is high, there are still proteins of low abundances beyond the detection limit of MS (PMID:23196971). Cp110 protein is one of such proteins beyond detection limit of MS. Because Cp110 is not detected, there is no TMT reporter ions for quantification of Cp110. Thus, Cp110 is not identified as a differential protein by MS, not due to discrepancy but due to low abundance in expression in spermatids.

10) Methods section: information describing how the CEP78 antibodies were produced is missing.

Thank the reviewer for the suggestion. We have added the following detail information of the antibody in the section of Materials and methods: the custom made human CEP78 and mouse Cep78 antibodies were produced using antigen sequence p457-741 of mouse Cep78 (NP_932136.2) and p100-515 of human CEP78 protein (NP_001092272.1) by Abclonal (Wuhan, China). The sequences were cloned to pET-28a (+) vectors, expressed in BL21 prokaryotic cells and immunized with experimental Japanese white rabbit, respectively. After the animals are sacrificed, the rabbit serum is subjected to antigen affinity purification. Which is at Page 15-16, line 341-348, line 357-358 in our new version of manuscript.

Reviewer #2 (Recommendations for the authors):1. I suggest rewriting the introduction section to make the statement accurate.

Thank the reviewer for the comment. We have rewritten the introduction section according to Reviewers’ suggestion to make it more clear and concise.

2. Better quantify the data as suggested in my specific comments.

Thank the reviewer for the suggestion. For Figure 4K, we stained *Cep78^+/-^* and *Cep78^-/-^* spermatids with anti-Centrin 1 to measure the centriole length. The statistical data of centriole length are now provided (Figure 4L), showing significantly increased centriole lengths in *Cep78^-/-^*spermatids.

For Figure 1F, we have quantified the immunofluorescent intensity of cone arrestin in light-adapted retinas of *Cep78^+/-^* and *Cep78^-/-^* mice at 3-month. The results indicate that immunofluorescent intensity of the cone arrestin was lower in *Cep78^-/-^* mice.

3. Conduct experiments suggested to provide evidence to support the conclusion. To show the three protein form timer, additional experimental evidence such as gel filtration is needed. To better visualize CC, NPHP1 or other proper markers should be used.

Thank the reviewer for the suggestions. According to the reviewer’s suggestion, we performed the gel filtration. And the results showed that Cep78, Ift20 and Ttc21a co-fractioned in a testicular protein complex at the size between158 kDa to 670 kDa (Figure 7E).

For CC visualization, we have stained anti-Nphp1 in retinal cryosections from *Cep78^+/-^* and *Cep78^-/-^* mice to visualize connecting cilium (Figure 2A-B). The results indicated that connecting cilia are shortened in *Cep78^-/-^* mice compared to *Cep78^+/-^*(Figure 2A-B). We also re-measured the length of CC photographed with TEM (Figure 2C-D). The ciliary structure out of outer segment is the CC, and is subjected to length measurement. The part inside the outlet segment is axoneme, which is not included in the length statistics. Consistent with our Nphp1 immunofluorescence. Our TEM results showed photoreceptors of *Cep78^-/-^* mice had shortened CC (Figures 2C-D).

Reviewer #3 (Recommendations for the authors):The experimental base is sufficient in this research to validate the statements. However, writing improvement and coherence in presentation would help to improve the readability significantly.

Thank the reviewer for the suggestion. We have carefully checked the language of the manuscript, we believe that our new version of the article can provide a better reading experience.

Following are suggestions for manuscript improvement:– It was not immediately clear if human mutation was identified by the authors and presented for the first time in this publication or was already done before. 'In this study, based on results of a male patient carrying CEP78 mutation and Cep78 gene knockout mice, we report CEP78 as a new causative gene for a distinct syndrome involving two phenotypes, CRD and male sterility[16].' I have followed reference 16 to understand that it is not related to discovery of CEP78 mutation. It is unclear why reference 16 is included (lanes 106-108).

Thank the reviewer for pointing this out. Reference 16 was not correctly cited, during revision we have removed this citation. We apologize for such mistake. The patient presented was the same as that previously reported by Fu et al. (10.1136/jmedgenet-2016-104166). To make this clear, we have added description of “We further explored whether c.1629-2A>G mutation, previously reported by Fu et al. [8], would disturb CEP78 protein expression and male fertility. ” at Page 29, Line 634-636 in the revised manuscript.

– Missing clear information if the patient/-s are the same as published in Qing Fu et al., (10.1136/jmedgenet-2016-104166).

Thank the reviewer for this question. The patient in our study is the same as published in Qing Fu et al. By citing Fu et al’s report when patient’s information was mentioned, we now clearly described that the patient was the same as published in Qing Fu et al. We added “We further explored whether c.1629-2A>G mutation, previously reported by Fu et al. 8[], would disturb CEP78 protein expression and male fertility.” in at Page 29, Line 634-636 in the revised version of the article.

– If the patient's CEP78 mutation is the same as published in Qing Fu et al., (10.1136/jmedgenet-2016-104166) why it was reported as causing Usher syndrome and in current the publication it is not mentioned at all? Do authors think that CEP78 mutation is associated with Usher's syndrome? It is important to discuss this.

Thank the reviewer for the comments. The patient studied in this manuscript is the same as that reported in publication by Qing Fu et al. (10.1136/jmedgenet-2016-104166), thus, we added “We have previously linked the *CEP78* c.1629-2A>G mutation to autosomal recessive CRDHL, and revealed a 10 bp deletion of *CEP78* exon 14 in mRNA of white blood cells from the patient carrying *CEP78* c.1629-2A>G mutation[8]. We further explored whether c.1629-2A>G mutation, previously reported by Fu et al. [8], would disturb CEP78 protein expression and male fertility.”

In Qing Fu et al. ’s study, we termed CEP78 mutation caused disease “a distinct type of Usher syndrome”, because patients with CEP78 mutation have syndromes similar but not identical to Usher syndrome. CEP78 disruption distinctively affects cone photoreceptors earlier, while in all other Usher syndrome types, rod photoreceptor functions are initially compromised (PMID: 11701652). This indicates CEP78 mutation is associated with a distinct type of Usher syndrome. So, with revisit of this patient, we found it is more accurate to describe the patient’s symptoms with “cone rod dystrophy”. So, in our new version of manuscript, we report this patient is associate with cone rod dystrophy. We hope our explanation can reduce the misunderstandings.

– The missense variant in CEP78 was shown to be associated with astenoteratozoospermia and reduced male fertility but, no morphological defects were found in sperm (Ascari, G., et al., 2020). Authors should consider discussing this obvious phenotypic and functional difference between missense variant and 10 bp deletion in CEP78, that they have studied. Link to IP-MS data submitted to Dryad, Dataset, (https://doi.org/10.5061/dryad.6djh9w12z) was not working during the revision process (Error message: link incorrect or not activated yet).

Thanks for the comment.

In this study, we reported a patient with *CEP78* c.1629-2A>G mutation, which caused homozygous 10 bp deletion of CEP78 exon 14 in mRNA, and CEP78 protein absence. Mutation of CEP78 caused complete absence of CEP78 protein. We found this patient has male infertility and multiple morphological abnormalities of the sperm flagella (MMAF) with oligoasthenoteratozoospermia.

Ascari, G., et al. reported 3 patients with male reproductive phenotypes, among whom only one patient had mutations only in *CEP78* gene and absence of its full length protein expression. The patient had compound heterozygous mutations with a missense and a nonsense mutation (c.[449T>C];[1462-1G>T] p.[Leu150Ser];[?]), and exhibited asthenoteratozoospermia and male infertility, with the sperm phenotype less severe than that observed in our study. The patient in our study showed phenotype of oligoasthenoteratozoospermia with MMAF, consistent with the phenotype observed in Cep78 knockout mice. Whether the patient reported by Ascari, G., et al. had truncated CEP78 by nonsense mutation was not analyzed, which might contribute the phenotypic differences between patients.

For the link to IP-MS data, we have renewed the link of IP-MS data as https://datadryad.org/stash/share/kCnVtra37ZZ2zfP0_CtfK9czhA6oN9neoS11KNILdtE

The updated link is accessible now, it’s at Page 40 Line 864-873 in the revised manuscript.

– What part of the CEP78 protein antibody detects (before or after 10 bp deletion)? This would be interesting to know in order to eliminate the possibility of the existence of the truncated CER78 protein in the affected patients.

Thanks for reviewer’s comments. CEP78 antibody was generated based on the antigen sequence of p100-515 of CEP78 protein (NP_001092272.1). And the antigen sequence is before the 10 bp deletion in the patient with mutation of canonical splicing acceptor site of exon 14 (c.1629-2A>G, p.?). If truncated CEP78 p.G545Pfs*6 protein is expressed, it will contain the antigen sequence and is expected to be detected by our anti-CEP78 antibody. Our Western blot results indicated that no truncated CEP78 could be detected.

We have added the following detail information of the antibody in the section of Materials and methods: the custom made human CEP78 and mouse Cep78 antibodies were produced using antigen sequence p457-741 of mouse Cep78 (NP_932136.2) and p100-515 of human CEP78 protein (NP_001092272.1) by Abclonal (Wuhan, China). The sequences were cloned to pET-28a (+) vectors, expressed in BL21 prokaryotic cells and immunized with experimental Japanese white rabbit, respectively. After the animals are sacrificed, the rabbit serum is subjected to antigen affinity purification. Which is at Page 15-16, line 341-348, line 357-358, in our new version of manuscript.

– Authors refer to the same publication but use different reference numbers in the text and in the reference list: reference: nr 8 Fu, Q., et al., CEP78 is mutated in a distinct type of Usher syndrome. J Med Genet, 2017. 54(3): p. 190-195. nr 23: Fu, Q., et al., CEP78 is mutated in a distinct type of Usher syndrome. J Med Genet, 2017. 54(3): p. 190-195. Please address this.

Thank the reviewer for the reminding. We have removed identical reference 23 in the revised manuscript.

– Gene should be italic: lines 153, 177, 227, 233, 240, 246, 273, 274,278, 279 and oth.

We are grateful to the comments. According to the suggestion, we have changed the above-mentioned gene names to font italic at lines 456, 481, 535, 541, 548, 554, 580, 581, and 584 in the revised manuscript.

– Line 271; deletion of gene not protein.

Thanks for the comment. We have italicized the gene name *Cep78* in Page 26, Line 578 in the revised manuscript.

– English language revision would improve readability.

Thank you for this suggestion, we have carefully checked the language writing, we believe that our revised version of the manuscript will give a better reading experience.